# Fast Exact Leverage Score Sampling from Khatri-Rao Products with Applications to Tensor Decomposition

**Vivek Bharadwaj [1,2], Osman Asif Malik [2], Riley Murray [3,2,1],**
**Laura Grigori [4], Aydın Buluç [2,1], James Demmel [1]**
[1]Electrical Engineering and Computer Science Department, UC Berkeley
[2]Computational Research Division, Lawrence Berkeley National Lab
[3]International Computer Science Institute
[4] Institute of Mathematics, EPFL & Lab for Simulation and Modelling, Paul Scherrer Institute

## Abstract

We present a data structure to randomly sample rows from the Khatri-Rao product of several matrices according to the exact distribution of its leverage scores. Our proposed sampler draws each row in time logarithmic in the height of the Khatri-Rao product and quadratic in its column count, with persistent space overhead at most the size of the input matrices. As a result, it tractably draws samples even when the matrices forming the Khatri-Rao product have tens of millions of rows each. When used to sketch the linear least squares problems arising in CANDECOMP / PARAFAC tensor decomposition, our method achieves lower asymptotic complexity per solve than recent state-of-the-art methods. Experiments on billion-scale sparse tensors validate our claims, with our algorithm achieving higher accuracy than competing methods as the decomposition rank grows.

## 1 Introduction

The Khatri-Rao product (KRP, denoted by $\odot$) is the column-wise Kronecker product of two matrices, and it appears in diverse applications across numerical analysis and machine learning [16]. We examine overdetermined linear least squares problems of the form $\min_X \|AX - B\|_F$, where the design matrix $A = U_1 \odot ... \odot U_N$ is the Khatri-Rao product of matrices $U_j \in \mathbb{R}^{I_j \times R}$. These problems appear prominently in signal processing [23], compressed sensing [31], inverse problems related to partial differential equations [5], and alternating least squares (ALS) CANDECOMP / PARAFAC (CP) tensor decomposition [13]. In this work, we focus on the case where $A$ has moderate column count (several hundred at most). Despite this, the problem remains formidable because the height of $A$ is $\prod_{j=1}^{N} I_j$. For row counts $I_j$ in the millions, it is intractable to even materialize $A$ explicitly.

Several recently-proposed randomized sketching algorithms can approximately solve least squares problems with Khatri-Rao product design matrices [4, 12, 15, 18, 29]. These methods apply a sketching operator $S$ to the design and data matrices to solve the reduced least squares problem $\min_{\tilde{X}} \left\| SA\tilde{X} - SB \right\|_F$, where $S$ has far fewer rows than columns. For appropriately chosen $S$, the residual of the downsampled system falls within a specified tolerance $\varepsilon$ of the optimal residual with high probability $1 - \delta$. In this work, we constrain $S$ to be a *sampling matrix* that selects and reweights a subset of rows from both $A$ and $B$. When the rows are selected according to the distribution of *statistical leverage scores* on the design matrix $A$, only $\tilde{O}\left(R/(\varepsilon\delta)\right)$ samples are required (subject to the assumptions at the end of section 2.1). The challenge, then, is to efficiently sample according to the leverage scores when $A$ has Khatri-Rao structure.

37th Conference on Neural Information Processing Systems (NeurIPS 2023).

We propose a leverage-score sampler for the Khatri-Rao product of matrices with tens of millions of rows each. After construction, our sampler draws each row in time quadratic in the column count, but logarithmic in the total row count of the Khatri-Rao product. Our core contribution is the following theorem.

**Theorem 1.1** (Efficient Khatri-Rao Product Leverage Sampling). *Given $U_1, ..., U_N$ with $U_j \in \mathbb{R}^{I_j \times R}$, there exists a data structure satisfying the following:*

1. *The data structure has construction time $O\left(\sum_{j=1}^{N} I_j R^2\right)$ and requires additional storage space $O\left(\sum_{j=1}^{N} I_j R\right)$. If a single entry in a matrix $U_j$ changes, it can be updated in time $O(R \log(I_j/R))$. If the entire matrix $U_j$ changes, it can be updated in time $O\left(I_j R^2\right)$.*

2. *The data structure produces $J$ samples from the Khatri-Rao product $U_1 \odot ... \odot U_N$ according to the exact leverage score distribution on its rows in time*

$$O\left(NR^3 + J \sum_{k=1}^{N} R^2 \log \max(I_k, R)\right)$$

*using $O(R^3)$ scratch space. The structure can also draw samples from the Khatri-Rao product of any subset of $U_1, ..., U_N$.*

The efficient update property and ability to exclude one matrix are important in CP decomposition. When the inputs $U_1, ..., U_N$ are sparse, an analogous data structure with $O\left(R \sum_{j=1}^{N} \text{nnz}(U_j)\right)$ construction time and $O\left(\sum_{j=1}^{N} \text{nnz}(U_j)\right)$ storage space exists with identical sampling time. Since the output factor matrices $U_1, ..., U_N$ are typically dense, we defer the proof to Appendix A.8. Combined with error guarantees for leverage-score sampling, we achieve an algorithm for alternating least squares CP decomposition with asymptotic complexity lower than recent state-of-the-art methods (see Table 1).

Our method provides the most practical benefit on sparse input tensors, which may have dimension lengths in the tens of millions (unlike dense tensors that quickly incur intractable storage costs at large dimension lengths) [25]. On the Amazon and Reddit tensors with billions of nonzero entries, our algorithm STS-CP can achieve 95% of the fit of non-randomized ALS between 1.5x and 2.5x faster than a high-performance implementation of the state-of-the-art CP-ARLS-LEV algorithm [15]. Our algorithm is significantly more sample-efficient; on the Enron tensor, only $\sim 65,000$ samples per solve were required to achieve the 95% accuracy threshold above a rank of 50, which could not be achieved by CP-ARLS-LEV with even 54 times as many samples.

Table 1: Asymptotic Complexity to decompose an $N$-dimensional $I \times ... \times I$ dense tensor via CP alternating least squares. For randomized algorithms, each approximate least-squares solution has residual within $(1 + \varepsilon)$ of the optimal value with high probability $1 - \delta$. Factors involving $\log R$ and $\log(1/\delta)$ are hidden ($\tilde{O}$ notation). See A.1 for details.

| Algorithm | Complexity per Iteration |
| --- | --- |
| CP-ALS [13] | $N(N+I)I^{N-1}R$ |
| CP-ARLS-LEV [15] | $N(R+I)R^N/(\varepsilon\delta)$ |
| TNS-CP [19] | $N^3 I R^3/(\varepsilon\delta)$ |
| Gaussian TNE [17] | $N^2(N^{1.5}R^{3.5}/\varepsilon^3 + IR^2)/\varepsilon^2$ |
| **STS-CP (ours)** | $N(NR^3 \log I + IR^2)/(\varepsilon\delta)$ |

## 2 Preliminaries and Related Work

**Notation.** We use $[N]$ to denote the set $\{1, ..., N\}$ for a positive integer $N$. We use $\tilde{O}$ notation to indicate the presence of multiplicative terms polylogarithmic in $R$ and $(1/\delta)$ in runtime complexities. For the complexities of our methods, these logarithmic factors are no more than $O(\log(R/\delta))$. We

use Matlab notation $A[i,:], A[:,i]$ to index rows, resp. columns, of matrices. For consistency, we use the convention that $A[i,:]$ is a row vector. We use $\cdot$ for standard matrix multiplication, $\circledast$ as the elementwise product, $\otimes$ to denote the Kronecker product, and $\odot$ for the Khatri-Rao product. See Appendix A.2 for a definition of each operation. Given matrices $A \in \mathbb{R}^{m_1 \times n}, B \in \mathbb{R}^{m_2 \times n}$, the $j$-th column of the Khatri-Rao product $A \odot B \in \mathbb{R}^{m_1 m_2 \times n}$ is the Kronecker product $A[:,j] \otimes B[:,j]$.

We use angle brackets $\langle \cdot, ..., \cdot \rangle$ to denote a **generalized inner product**. For identically-sized vectors / matrices, it returns the sum of all entries in their elementwise product. For $A, B, C \in \mathbb{R}^{m \times n}$,

$$\langle A, B, C \rangle := \sum_{i=1, j=1}^{m,n} A[i,j]\, B[i,j]\, C[i,j].$$

Finally, $M^+$ denotes the pseudoinverse of matrix $M$.

## 2.1 Sketched Linear Least Squares

A variety of random sketching operators $S$ have been proposed to solve overdetermined least squares problems $\min_X \|AX - B\|_F$ when $A$ has no special structure [30, 2]. When $A$ has Khatri-Rao product structure, prior work has focused on *sampling* matrices [6, 15], which have a single nonzero entry per row, operators composed of fast Fourier / trigonometric transforms [12], or Countsketch-type operators [27, 1]. For tensor decomposition, however, the matrix $B$ may be sparse or implicitly specified as a black-box function. When $B$ is sparse, Countsketch-type operators still require the algorithm to iterate over all nonzero values in $B$. As Larsen and Kolda [15] note, operators similar to the FFT induce fill-in when applied to a sparse matrix $B$, destroying the benefits of sketching. Similar difficulties arise when $B$ is implicitly specified. This motivates our decision to focus on row sampling operators, which only touch a subset of entries from $B$. Let $\hat{x}_1, ..., \hat{x}_J$ be a selection of $J$ indices for the rows of $A \in \mathbb{R}^{I \times R}$, sampled i.i.d. according to a probability distribution $q_1, ..., q_I$. The associated sampling matrix $S \in \mathbb{R}^{J \times I}$ is specified by

$$S[j,i] = \begin{cases} \frac{1}{\sqrt{Jq_i}}, & \text{if } \hat{x}_j = i \\ 0, & \text{otherwise} \end{cases}$$

where the weight of each nonzero entry corrects bias induced by sampling. When the probabilities $q_j$ are proportional to the *leverage scores* of the rows of $A$, strong guarantees apply to the solution of the downsampled problem.

**Leverage Score Sampling.** The leverage scores of a matrix assign a measure of importance to each of its rows. The leverage score of row $i$ from matrix $A \in \mathbb{R}^{I \times R}$ is given by

$$\ell_i = A[i,:]\,(A^\top A)^+ A[i,:]^\top \tag{1}$$

for $1 \leq i \leq I$. Leverage scores can be expressed equivalently as the squared row norms of the matrix $Q$ in any reduced $QR$ factorization of $A$ [8]. The sum of all leverage scores is the rank of $A$ [30]. Dividing the scores by their sum, we induce a probability distribution on the rows used to generate a sampling matrix $S$. The next theorem has appeared in several works, and we take the form given by Malik et al. [19]. For an appropriate sample count, it guarantees that the residual of the downsampled problem is close to the residual of the original problem.

**Theorem 2.1** (Guarantees for Leverage Score Sampling). *Given $A \in \mathbb{R}^{I \times R}$ and $\varepsilon, \delta \in (0, 1)$, let $S \in \mathbb{R}^{J \times I}$ be a leverage score sampling matrix for $A$. Further define $\tilde{X} = \arg\min_X \|SAX - SB\|_F$. If $J \gtrsim R \max(\log(R/\delta), 1/(\varepsilon\delta))$, then with probability at least $1 - \delta$ it holds that*

$$\left\| A\tilde{X} - B \right\|_F \leq (1 + \varepsilon) \min_X \|AX - B\|_F.$$

For the applications considered in this work, $R$ ranges up to a few hundred. As $\varepsilon$ and $\delta$ tend to 0 with fixed $R$, $1/(\varepsilon\delta)$ dominates $\log(R/\delta)$. Hence, we assume that the minimum sample count $J$ to achieve the guarantees of the theorem is $\Omega(R/(\varepsilon\delta))$.

## 2.2 Prior Work

**Khatri-Rao Product Leverage Score Sampling.** Well-known sketching algorithms exist to quickly estimate the leverage scores of dense matrices [8]. These algorithms are, however, intractable for $A = U_1 \odot ... \odot U_N$ due to the height of the Khatri-Rao product. Cheng et al. [6] instead approximate each score as a product of leverage scores associated with each matrix $U_j$. Larsen and Kolda [15] propose CP-ARLS-LEV, which uses a similar approximation and combines random sampling with a deterministic selection of high-probability indices. Both methods were presented in the context of CP decomposition. To sample from the Khatri-Rao product of $N$ matrices, both require $O(R^N/(\varepsilon\delta))$ samples to achieve the $(\varepsilon, \delta)$ guarantee on the residual of each least squares solution. These methods are simple to implement and perform well when the Khatri-Rao product has column count up to 20-30. On the other hand, they suffer from high sample complexity as $R$ and $N$ increase. The TNS-CP algorithm by Malik et al. [19] samples from the exact leverage score distribution, thus requiring only $O(R/(\varepsilon\delta))$ samples per least squares solve. Unfortunately, it requires time $O\left(\sum_{j=1}^{N} I_j R^2\right)$ to draw each sample.

**Comparison to Woodruff and Zandieh.** The most comparable results to ours appear in work by Woodruff and Zandieh [29], who detail an algorithm for approximate ridge leverage-score sampling for the Khatri-Rao product in near input-sparsity time. Their work relies on a prior oblivious method by Ahle et al. [1], which sketches a Khatri-Rao product using a sequence of Countsketch / OSNAP operators arranged in a tree. Used in isolation to solve a linear least squares problem, the tree sketch construction time scales as $O\left(\frac{1}{\varepsilon}\sum_{j=1}^{N} \text{nnz}(U_j)\right)$ and requires an embedding dimension quadratic in $R$ to achieve the $(\varepsilon, \delta)$ solution-quality guarantee. Woodruff and Zandieh use a collection of these tree sketches, each with carefully-controlled approximation error, to design an algorithm with linear runtime dependence on the column count $R$. On the other hand, the method exhibits $O(N^7)$ scaling in the number of matrices involved, has $O(\varepsilon^{-4})$ scaling in terms of the desired accuracy, and relies on a sufficiently high ridge regularization parameter. Our data structure instead requires construction time quadratic in $R$. In exchange, we use distinct methods to design an efficiently-updatable sampler with runtime linear in both $N$ and $\varepsilon^{-1}$. These properties are attractive when the column count $R$ is below several thousand and when error as low as $\epsilon \approx 10^{-3}$ is needed in the context of an iterative solver (see Figure 5). Moreover, the term $O(R^2 \sum_{j=1}^{N} I_j)$ in our construction complexity arises from symmetric rank-$k$ updates, a highly-optimized BLAS3 kernel on modern CPU and GPU architectures. Appendix A.3 provides a more detailed comparison between the two approaches.

**Kronecker Regression.** Kronecker regression is a distinct (but closely related) problem to the one we consider. There, $A = U_1 \otimes ... \otimes U_N$ and the matrices $U_i$ have potentially distinct column counts $R_1, ..., R_N$. While the product distribution of leverage scores from $U_1, ..., U_N$ provides only an approximation to the leverage score distribution of the Khatri-Rao product [6, 15], it provides the *exact* leverage distribution for the Kronecker product. Multiple works [7, 9] combine this property with other techniques, such as dynamically-updatable tree-sketches [21], to produce accurate and updatable Kronecker sketching methods. None of these results apply directly in our case due to the distinct properties of Kronecker and Khatri-Rao products.

## 3 An Efficient Khatri-Rao Leverage Sampler

Without loss of generality, we will prove part 2 of Theorem 1.1 for the case where $A = U_1 \odot ... \odot U_N$; the case that excludes a single matrix follows by reindexing matrices $U_k$. We further assume that $A$ is a nonzero matrix, though it may be rank-deficient. Similar to prior sampling works [18, 29], our algorithm will draw one sample from the Khatri-Rao product by sampling a row from each of $U_1, U_2, ....$ in sequence and computing their Hadamard product, with the draw from $U_j$ conditioned on prior draws from $U_1, ..., U_{j-1}$.

Let us index each row of $A$ by a tuple $(i_1, ..., i_N) \in [I_1] \times ... \times [I_N]$. Equation (1) gives

$$\ell_{i_1,...,i_N} = A\left[(i_1, ..., i_N), :\right] (A^\top A)^+ A\left[(i_1, ..., i_N), :\right]^\top. \tag{2}$$

For $1 \leq k \leq N$, define $G_k := U_k^\top U_k \in \mathbb{R}^{R \times R}$ and $G := \left(\circledast_{k=1}^{N} G_k\right) \in \mathbb{R}^{R \times R}$; it is a well-known fact that $G = A^\top A$ [13]. For a single row sample from $A$, let $\hat{s}_1, ..., \hat{s}_N$ be random variables for the

draws from multi-index set $[I_1] \times ... \times [I_N]$ according to the leverage score distribution. Assume, for some $k$, that we have already sampled an index from each of $[I_1], ..., [I_{k-1}]$, and that the first $k-1$ random variables take values $\hat{s}_1 = s_1, ..., \hat{s}_{k-1} = s_{k-1}$. We abbreviate the latter condition as $\hat{s}_{<k} = s_{<k}$. To sample from $I_k$, we seek the distribution of $\hat{s}_k$ conditioned on $\hat{s}_1, ...\hat{s}_{k-1}$. Define $h_{<k}$ as the transposed elementwise product[1] of rows already sampled:

$$h_{<k} := \overset{k-1}{\underset{i=1}{\circledast}} U_i [s_i, :]^\top. \tag{3}$$

Also define $G_{>k}$ as

$$G_{>k} := G^+ \circledast \overset{N}{\underset{i=k+1}{\circledast}} G_i. \tag{4}$$

Then the following theorem provides the conditional distribution of $\hat{s}_k$.

**Theorem 3.1** (Malik 2022, [18], Adapted). *For any $s_k \in [I_k]$,*

$$p(\hat{s}_k = s_k \mid \hat{s}_{<k} = s_{<k}) = C^{-1} \langle h_{<k} h_{<k}^\top, U_k [s_k, :]^\top U_k [s_k, :], G_{>k} \rangle$$
$$:= q_{h_{<k}, U_k, G_{>k}} [s_k] \tag{5}$$

*where $C = \langle h_{<k} h_{<k}^\top, U_k^\top U_k, G_{>k} \rangle$ is nonzero.*

We include the derivation of Theorem 3.1 from Equation (2) in Appendix A.4. Computing all entries of the probability vector $q_{h_{<k}, U_k, G_{>k}}$ would cost $O(I_j R^2)$ per sample, too costly when $U_j$ has millions of rows. It is likewise intractable (in preprocessing time and space complexity) to precompute probabilities for every possible conditional distribution on the rows of $U_j$, since the conditioning random variable has $\prod_{k=1}^{j-1} I_k$ potential values. Our key innovation is a data structure to sample from a discrete distribution of the form $q_{h_{<k}, U_k, G_{>k}}$ *without* materializing all of its entries or incurring superlinear cost in either $N$ or $\varepsilon^{-1}$. We introduce this data structure in the next section and will apply it twice in succession to get the complexity in Theorem 1.1.

### 3.1 Efficient Sampling from $q_{h, U, Y}$

We introduce a slight change of notation in this section to simplify the problem and generalize our sampling lemma. Let $h \in \mathbb{R}^R$ be a vector and let $Y \in \mathbb{R}^{R \times R}$ be a positive semidefinite (p.s.d.) matrix, respectively. Our task is to sample $J$ rows from a matrix $U \in \mathbb{R}^{I \times R}$ according to the distribution

$$q_{h, U, Y} [s] := C^{-1} \langle hh^\top, U^\top [s, :] U [s, :], Y \rangle \tag{6}$$

provided the normalizing constant $C = \langle hh^\top, U^\top U, Y \rangle$, is nonzero. We impose that all $J$ rows are drawn with the same matrices $Y$ and $U$, but potentially distinct vectors $h$. The following lemma establishes that an efficient sampler for this problem exists.

**Lemma 3.2** (Efficient Row Sampler). *Given matrices $U \in \mathbb{R}^{I \times R}, Y \in \mathbb{R}^{R \times R}$ with $Y$ p.s.d., there exists a data structure parameterized by positive integer $F$ that satisfies the following:*

1. *The structure has construction time $O(IR^2)$ and storage requirement $O(R^2 \lceil I/F \rceil)$. If $I < F$, the storage requirement drops to $O(1)$.*

2. *After construction, the data structure can produce a sample according to the distribution $q_{h, U, Y}$ in time $O(R^2 \log \lceil I/F \rceil + FR^2)$ for any vector $h$.*

3. *If $Y$ is a rank-1 matrix, the time per sample drops to $O(R^2 \log \lceil I/F \rceil + FR)$.*

This data structure relies on an adaptation of a classic binary-tree inversion sampling technique [22]. Consider a partition of the interval $[0, 1]$ into $I$ bins, the $i$-th having width $q_{h, U, Y} [i]$. We sample $d \sim \text{Uniform} [0, 1]$ and return the index of the containing bin. We locate the bin index through a binary search terminated when at most $F$ bins remain in the search space, which are then scanned in linear time. Here, $F$ is a tuning parameter that we will use to control sampling complexity and space usage.

---

[1] For $a > b$, assume that $\overset{b}{\underset{i=a}{\circledast}} (...)$ produces a vector / matrix filled with ones.

We can regard the binary search as a walk down a full, complete binary tree $T_{I,F}$ with $\lceil I/F \rceil$ leaves, the nodes of which store contiguous, disjoint segments $S(v) = \{S_0(v)..S_1(v)\} \subseteq [I]$ of size at most $F$. The segment of each internal node is the union of segments held by its children, and the root node holds $\{1, ..., I\}$. Suppose that the binary search reaches node $v$ with left child $L(v)$ and maintains the interval $[\text{low}, \text{high}] \subseteq [0, 1]$ as the remaining search space to explore. Then the search branches left in the tree iff $d < \text{low} + \sum_{i \in S(L(v))} q_{h,U,Y}[i]$.

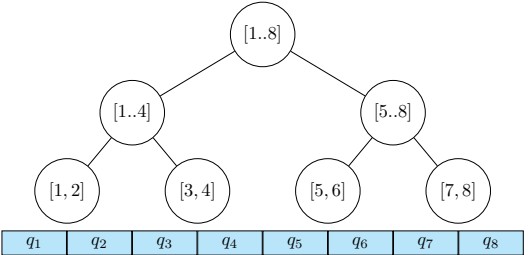

Figure 1: A segment tree $T_{8,2}$ and probability distribution $\{q_1, ..., q_8\}$ on $[1, ..., 8]$.

This branching condition can be evaluated efficiently if appropriate information is stored at each node of the segment tree. Excluding the offset "low", the branching threshold takes the form

$$\sum_{i \in S(v)} q_{h,U,Y}[i] = C^{-1} \langle hh^\top, \sum_{i \in S(v)} U[i,:]^\top U[i,:], Y \rangle := C^{-1} \langle hh^\top, G^v, Y \rangle. \tag{7}$$

Here, we call each matrix $G^v \in \mathbb{R}^{R \times R}$ a *partial Gram matrix*. In time $O(IR^2)$ and space $O(R^2 \lceil I/F \rceil)$, we can compute and cache $G^v$ for each node of the tree to construct our data structure. Each subsequent binary search costs $O(R^2)$ time to evaluate Equation (7) at each of $\log \lceil I/F \rceil$ internal nodes and $O(FR^2)$ to evaluate $q_{h,U,Y}$ at the $F$ indices held by each leaf, giving point 2 of the lemma. This cost at each leaf node reduces to $O(FR)$ in case $Y$ is rank-1, giving point 3. A complete proof of this lemma appears in Appendix A.5.

## 3.2 Sampling from the Khatri-Rao Product

We face difficulties if we directly apply Lemma 3.2 to sample from the conditional distribution in Theorem 3.1. Because $G_{>k}$ is not rank-1 in general, we must use point 2 of the lemma, where no selection of the parameter $F$ allows us to simultaneously satisfy the space and runtime constraints of Theorem 1.1. Selecting $F = R$ results in cost $O(R^3)$ per sample (violating the runtime requirement in point 2), whereas $F = 1$ results in a superlinear storage requirement $O(IR^2)$ (violating the space requirement in point 1, and becoming prohibitively expensive for $I \geq 10^6$). To avoid these extremes, we break the sampling procedure into two stages. The first stage selects a 1-dimensional subspace spanned by an eigenvector of $G_{>k}$, while the second samples according to Theorem 3.1 after projecting the relevant vectors onto the selected subspace. Lemma 3.2 can be used for *both* stages, and the second stage benefits from point 3 to achieve better time and space complexity.

Below, we abbreviate $q = q_{h_{<k}, U_k, G_{>k}}$ and $h = h_{<k}$. When sampling from $I_k$, observe that $G_{>k}$ is the same for all samples. We compute a symmetric eigendecomposition $G_{>k} = V \Lambda V^\top$, where each column of $V$ is an eigenvector of $G_{>k}$ and $\Lambda = \text{diag}((\lambda_u)_{u=1}^R)$ contains the eigenvalues along the diagonal. This allows us to rewrite entries of $q$ as

$$q[s] = C^{-1} \sum_{u=1}^R \lambda_u \langle hh^\top, U_k[s,:]^\top U_k[s,:], V[:,u] V[:,u]^\top \rangle. \tag{8}$$

Define matrix $W \in \mathbb{R}^{I_k \times R}$ elementwise by

$$W[t,u] := \langle hh^\top, U_k[t,:]^\top U_k[t,:], V[:,u] V[:,u]^\top \rangle$$

and observe that all of its entries are nonnegative. Since $\lambda_u \geq 0$ for all $u$ ($G_{>k}$ is p.s.d.), we can write $q$ as a mixture of probability distributions given by the normalized columns of $W$:

$$q = \sum_{u=1}^R w[u] \frac{W[:,u]}{\|W[:,u]\|_1},$$

where the vector $w$ of nonnegative weights is given by $w[u] = (C^{-1} \lambda_u \|W[:,u]\|_1)$. Rewriting $q$ in this form gives us the two stage sampling procedure: first sample a component $u$ of the mixture according to the weight vector $w$, then sample an index in $\{1..I_k\}$ according to the probability vector

defined by $W[:,u]/\|W[:,u]\|_1$. Let $\hat{u}_k$ be a random variable distributed according to the probability mass vector $w$. We have, for $C$ taken from Theorem 3.1,

$$
\begin{aligned}
p(\hat{u}_k = u_k) &= C^{-1}\lambda_{u_k}\sum_{t=1}^{I_k} W[t,u_k] \\
&= C^{-1}\lambda_{u_k}\langle hh^\top, V[:,u_k]V[:,u_k]^\top, G_k\rangle \\
&= q_{h,\sqrt{\Lambda}V^\top,G_k}[u_k].
\end{aligned}
\tag{9}
$$

Hence, we can use point 2 of Lemma 3.2 to sample a value for $\hat{u}_k$ efficiently. Because $\sqrt{\Lambda}V^\top$ has only $R$ rows with $R\sim 10^2$, we can choose tuning parameter $F=1$ to achieve lower time per sample while incurring a modest $O(R^3)$ space overhead. Now, introduce a random variable $\hat{t}_k$ with distribution conditioned on $\hat{u}_k = u_k$ given by

$$
p(\hat{t}_k = t_k \mid \hat{u}_k = u_k) := W[t_k,u_k]/\|W[:,u_k]\|_1.
\tag{10}
$$

This distribution is well-defined, since we suppose that $\hat{u}_k = u_k$ occurs with nonzero probability $e[u_k]$, which implies that $\|W[:,u_k]\|_1 \neq 0$. Our remaining task is to efficiently sample from the distribution above. Below, we abbreviate $\tilde{h} = V[:,u_k]\circledast h$ and derive

$$
\begin{aligned}
p(\hat{t}_k = t_k \mid \hat{u}_k = u_k) &= \frac{\langle hh^\top, U_k[t_k,:]^\top U_k[t_k,:], V[:,u_k]V[:,u_k]^\top\rangle}{\|W[:,u_k]\|_1} \\
&= \frac{\langle \tilde{h}\tilde{h}^\top, U_k[t_k,:]^\top U_k[t_k,:], [1]\rangle}{\|W[:,u_k]\|_1} \\
&= q_{\tilde{h},U_k,[1]}[t_k].
\end{aligned}
\tag{11}
$$

Based on the last line of Equation (11), we apply Lemma 3.2 again to build an efficient data structure to sample a row of $U_k$. Since $Y=[1]$ is a rank-1 matrix, we can use point 3 of the lemma and select a larger parameter value $F=R$ to reduce space usage. The sampling time for this stage becomes $O(R^2\log\lceil I_j/R\rceil)$.

To summarize, Algorithms 1 and 2 give the construction and sampling procedures for our data structure. They rely on the "BuildSampler" and "RowSample" procedures from Algorithms 3 and 4 in Appendix A.5, which relate to the data structure in Lemma 3.2. In the construction phase, we build $N$ data structures from Lemma 3.2 for the distribution in Equation (11). Construction costs $O\left(\sum_{j=1}^N I_j R^2\right)$, and if any matrix $U_j$ changes, we can rebuild $Z_j$ in isolation. Because $F=R$, the space required for $Z_j$ is $O(I_j R)$. In the sampling phase, the procedure in Algorithm 2 accepts an optional index $j$ of a matrix to exclude from the Khatri-Rao product. The procedure begins by computing the symmetric eigendecomposition of each matrix $G_{>k}$. The eigendecomposition is computed only once per binary tree structure, and its computation cost is amortized over all $J$ samples. It then creates data structures $E_k$ for each of the distributions specified by Equation (9). These data structures (along with those from the construction phase) are used to draw $\hat{u}_k$ and $\hat{t}_k$ in succession. The random variables $\hat{t}_k$ follow the distribution in Theorem 3.1 conditioned on prior draws, so the multi-index $(\hat{t}_k)_{k\neq j}$ follows the leverage score distribution on $A$, as desired. Appendix A.6 proves the complexity claims in the theorem and provides further details about the algorithms.

### 3.3 Application to Tensor Decomposition

A tensor is a multidimensional array, and the CP decomposition represents a tensor as a sum of outer products [13]. See Appendix A.9 for an overview. To approximately decompose tensor $\mathcal{T} \in \mathbb{R}^{I_1\times\cdots\times I_N}$, the popular alternating least squares (ALS) algorithm begins with randomly initialized factor matrices $U_j$, $U_j \in \mathbb{R}^{I_j\times R}$ for $1\leq j\leq N$. We call the column count $R$ the **rank** of the decomposition. Each round of ALS solves $N$ overdetermined least squares problems in sequence, each optimizing a single factor matrix while holding the others constant. The $j$-th least squares problem occurs in the update

$$
U_j := \arg\min_X \left\|U_{\neq j}\cdot X^\top - \mathrm{mat}(\mathcal{T},j)^\top\right\|_F
$$

where $U_{\neq j} = U_N\odot\ldots\odot U_{j+1}\odot U_{j-1}\odot\ldots\odot U_1$ is the Khatri-Rao product of all matrices excluding $U_j$ and $\mathrm{mat}(\cdot)$ denotes the mode-$j$ matricization of tensor $\mathcal{T}$. Here, we reverse the order of matrices

in the Khatri-Rao product to match the ordering of rows in the matricized tensor (see Appendix A.9 for an explicit formula for the matricization). These problems are ideal candidates for randomized sketching [4, 12, 15], and applying the data structure in Theorem 1.1 gives us the **STS-CP** algorithm.

**Corollary 3.3** (STS-CP). *Suppose $\mathcal{T}$ is dense, and suppose we solve each least squares problem in ALS with a randomized sketching algorithm. A leverage score sampling approach as defined in section 2 guarantees that with $\tilde{O}(R/(\varepsilon\delta))$ samples per solve, the residual of each sketched least squares problem is within $(1 + \varepsilon)$ of the optimal residual with probability $(1 - \delta)$. The efficient sampler from Theorem 1.1 brings the complexity of ALS to*

$$\tilde{O}\left(\frac{\#it}{\varepsilon\delta} \cdot \sum_{j=1}^{N} \left(NR^3 \log I_j + I_j R^2\right)\right)$$

*where "$\#it$" is the number of ALS iterations, and with any term $\log I_j$ replaced by $\log R$ if $I_j < R$.*

The proof appears in Appendix A.9 and combines Theorem 1.1 with Theorem 2.1. STS-CP also works for sparse tensors and likely provides a greater advantage here than the dense case, as sparse tensors tend to have much larger mode size [25]. The complexity for sparse tensors depends heavily on the sparsity structure and is difficult to predict. Nevertheless, we expect a significant speedup based on prior works that use sketching to accelerate CP decomposition [6, 15].

## 4 Experiments

Experiments were conducted on CPU nodes of NERSC Perlmutter, an HPE Cray EX supercomputer, and our code is available at https://github.com/vbharadwaj-bk/fast_tensor_leverage.git. On tensor decomposition experiments, we compare our algorithms against the random and hybrid versions of CP-ARLS-LEV proposed by Larsen and Kolda [15]. These algorithms outperform uniform sampling and row-norm-squared sampling, achieving excellent accuracy and runtime relative to exact ALS. In contrast to TNS-CP and the Gaussian tensor network embedding proposed by Ma and Solomonik (see Table 1), CP-ARLS-LEV is one of the few algorithms that can practically

---

**Algorithm 1** ConstructKRPSampler($U_1, ..., U_N$)

1: **for** $j = 1..N$ **do**
2:    $Z_j := \text{BuildSampler}(U_j, F = R, [1])$
3:    $G_j := U_j^\top U_j$

---

**Algorithm 2** KRPSample($j, J$)

1: $G := \circledast_{k \neq j} G_k$
2: **for** $k \neq j$ **do**
3:    $G_{>k} := G^+ \circledast \circledast_{k=j+1}^{N} G_k$
4:    Decompose $G_{>k} = V_k \Lambda_k V_k^\top$
5:    $E_k := \text{BuildSampler}(\sqrt{\Lambda_k} \cdot V_k^\top, F = 1, G_k)$
6: **for** $d = 1..J$ **do**
7:    $h = [1, ..., 1]^\top$
8:    **for** $k \neq j$ **do**
9:      $\hat{u}_k := \text{RowSample}(E_k, h)$
10:      $\hat{t}_k := \text{RowSample}(Z_k, h \circledast (V_k[:, \hat{u}_k]))$
11:      $h \mathrel{*}= U_k[\hat{t}_k, :]$
12:    $s_d = (\hat{t}_k)_{k \neq j}$
13: **return** $s_1, ..., s_J$

---

decompose sparse tensors with mode sizes in the millions. In the worst case, CP-ARLS-LEV requires $\tilde{O}(R^{N-1}/(\varepsilon\delta))$ samples per solve for an $N$-dimensional tensor to achieve solution guarantees like those in Theorem 2.1, compared to $\tilde{O}(R/(\varepsilon\delta))$ samples required by STS-CP. Appendices A.10, A.11, and A.13 provide configuration details and additional results.

### 4.1 Runtime Benchmark

Figure 2 shows the time to construct our sampler and draw 50,000 samples from the Khatri-Rao product of i.i.d. Gaussian initialized factor matrices. We quantify the runtime impacts of varying $N$, $R$, and $I$. The asymptotic behavior in Theorem 1.1 is reflected in our performance measurements, with the exception of the plot that varies $R$. Here, construction becomes disproportionately cheaper than sampling due to cache-efficient BLAS3 calls during construction. Even when the full Khatri-Rao product has $\approx 3.78 \times 10^{22}$ rows (for $I = 2^{25}, N = 3, R = 32$), we require only 0.31 seconds on average for sampling (top plot, rightmost points).

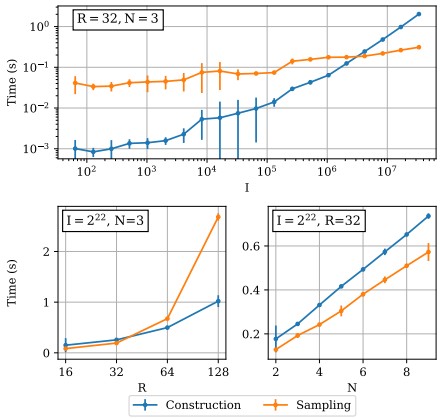

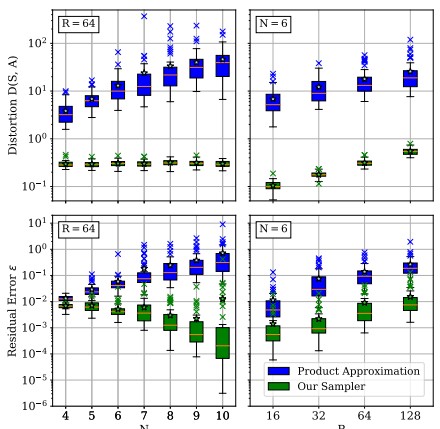

Figure 2: Average time (5 trials) to construct our proposed sampler and draw $J = 50,000$ samples from $U_1 \odot ... \odot U_N$, with $U_j \in \mathbb{R}^{I \times R} \, \forall j$. Error bars indicate 3 standard deviations.

Figure 3: Distortion and residual error (50 trials) for varying $R$ and $N$ on least squares, $I = 2^{16}, J = 5000$. "X" marks indicate outliers 1.5 times the interquartile range beyond the median, stars indicate means.

## 4.2 Least Squares Accuracy Comparison

We now test our sampler on least squares problems of the form $\min_x \|Ax - b\|$, where $A = U_1 \odot ... \odot U_N$ with $U_j \in \mathbb{R}^{I \times R}$ for all $j$. We initialize all matrices $U_j$ entrywise i.i.d. from a standard normal distribution and randomly multiply 1% of all entries by 10. We choose $b$ as a Kronecker product $c_1 \otimes ... \otimes c_N$, with each vector $c_j \in \mathbb{R}^I$ also initialized entrywise from a Gaussian distribution. We assume this form for $b$ to tractably compute the exact solution to the linear least squares problem and evaluate the accuracy of our randomized methods. We **do not** give our algorithms access to the Kronecker form of $b$; they are only permitted on-demand, black-box access to its entries.

For each problem instance, define the distortion of our sampling matrix $S$ with respect to the column space of $A$ as

$$D(S, A) = \frac{\kappa(SQ) - 1}{\kappa(SQ) + 1} \tag{12}$$

where $Q$ is an orthonormal basis for the column space of $A$ and $\kappa(SQ)$ is the condition number of $SQ$. A higher-quality sketch $S$ exhibits lower distortion, which quantifies the preservation of distances from the column space of $A$ to the column space of $SA$ [20]. For details about computing $\kappa(SQ)$ efficiently when $A$ is a Khatri-Rao product, see Appendix A.12. Next, define $\varepsilon = \frac{\text{residual}_{\text{approx}}}{\text{residual}_{\text{opt}}} - 1$, where residual$_{\text{approx}}$ is the residual of the randomized least squares algorithm. $\varepsilon$ is nonnegative and (similar to its role in Theorem 2.1) quantifies the quality of the randomized algorithm's solution.

For varying $N$ and $R$, Figure 3 shows the average values of $D$ and $\varepsilon$ achieved by our algorithm against the leverage product approximation used by Larsen and Kolda [15]. Our sampler exhibits nearly constant distortion $D$ for fixed rank $R$ and varying $N$, and it achieves $\varepsilon \approx 10^{-2}$ even when $N = 9$. The product approximation increases both the distortion and residual error by at least an order of magnitude.

## 4.3 Sparse Tensor Decomposition

We next apply STS-CP to decompose several large sparse tensors from the FROSTT collection [25] (see Appendix A.11 for more details on the experimental configuration). Our accuracy metric is the tensor fit. Letting $\tilde{\mathcal{T}}$ be our low-rank CP approximation, the fit with respect to ground-truth tensor $\mathcal{T}$ is $\text{fit}(\tilde{\mathcal{T}}, \mathcal{T}) = 1 - \left\|\tilde{\mathcal{T}} - \mathcal{T}\right\|_F / \|\mathcal{T}\|_F$.

Table 4 in Appendix A.13.1 compares the runtime per round of our algorithm against the Tensorly Python package [14] and Matlab Tensor Toolbox [3], with dramatic speedup over both. As Figure 4 shows, the fit achieved by CP-ARLS-LEV compared to STS-CP degrades as the rank increases for

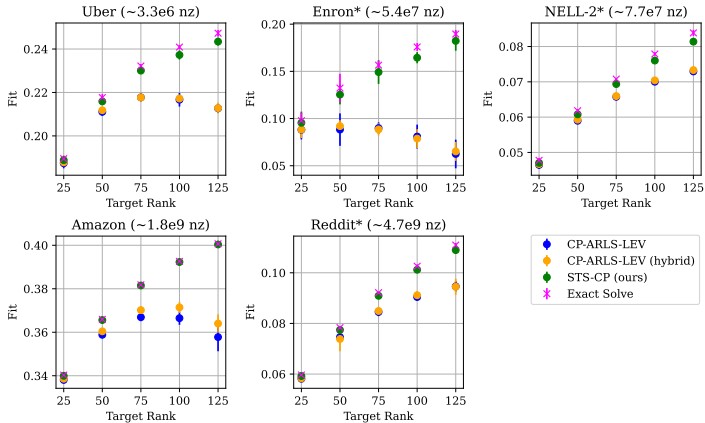

Figure 4: Average fits (8 trials) achieved by randomized ($J = 2^{16}$) and exact ALS for sparse tensor CP decomposition. Error bars indicate 3 standard deviations. See Appendix A.11 for details.

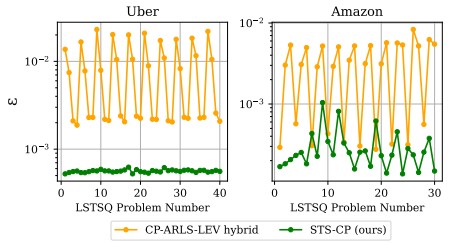

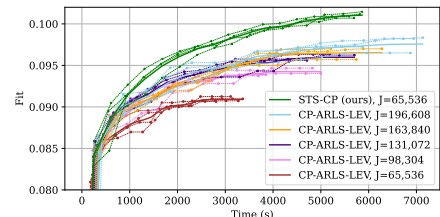

Figure 5: Average $\varepsilon$ (5 runs) for randomized least squares solves in 10 ALS rounds, $R = 50$.

Figure 6: Fit vs. time, Reddit tensor, $R = 100$. Thick lines are averages 4 trial interpolations.

fixed sample count. By contrast, STS-CP improves the fit consistently, with a significant improvement at rank 125 over CP-ARLS-LEV. Timings for both algorithms are available in Appendix A.13.5. Figure 5 explains the higher fit achieved by our sampler on the Uber and Amazon tensors. In the first 10 rounds of ALS, we compute the exact solution to each least squares problem before updating the factor matrix with a randomized algorithm's solution. Figure 5 plots $\varepsilon$ as ALS progresses for hybrid CP-ARLS-LEV and STS-CP. The latter consistently achieves lower residual per solve. We further observe that CP-ARLS-LEV exhibits an oscillating error pattern with period matching the number of modes $N$.

To assess the tradeoff between sampling time and accuracy, we compare the fit as a function of ALS update time for STS-CP and random CP-ARLS-LEV in Figure 6 (time to compute the fit excluded). On the Reddit tensor with $R = 100$, we compared CP-ARLS-LEV with $J = 2^{16}$ against CP-ARLS-LEV with progressively larger sample count. Even with $2^{18}$ samples per randomized least squares solve, CP-ARLS-LEV cannot achieve the maximum fit of STS-CP. Furthermore, STS-CP makes progress more quickly than CP-ARLS-LEV. See Appendix A.13.4 for similar plots for other datasets.

## 5 Discussion and Future Work

Our method for exact Khatri-Rao leverage score sampling enjoys strong theoretical guarantees and practical performance benefits. Especially for massive tensors such as Amazon and Reddit, our randomized algorithm's guarantees translate to faster progress to solution and higher final accuracies. The segment tree approach described here can be applied to sample from tensor networks besides the Khatri-Rao product. In particular, modifications to Lemma 3.2 permit efficient leverage sampling from a contraction of 3D tensor cores in ALS tensor train decomposition. We leave the generalization of our fast sampling technique as future work.

## Acknowledgements, Funding, and Disclaimers

We thank the referees for valuable feedback which helped improve the paper.

V. Bharadwaj was supported by the U.S. Department of Energy, Office of Science, Office of Advanced Scientific Computing Research, Department of Energy Computational Science Graduate Fellowship under Award Number DE-SC0022158. O. A. Malik and A. Buluç were supported by the Office of Science of the DOE under Award Number DE-AC02-05CH11231. L. Grigori was supported by the European Research Council (ERC) under the European Union's Horizon 2020 research and innovation program through grant agreement 810367. R. Murray was supported by Laboratory Directed Research and Development (LDRD) funding from Berkeley Lab, provided by the Director, Office of Science, of the U.S. DOE under Contract No. DE-AC02-05CH11231. R. Murray was also funded by an NSF Collaborative Research Framework under NSF Grant Nos. 2004235 and 2004763. This research used resources of the National Energy Research Scientific Computing Center, a DOE Office of Science User Facility, under Contract No. DE-AC02-05CH11231 using NERSC award ASCR-ERCAP0024170.

This report was prepared as an account of work sponsored by an agency of the United States Government. Neither the United States Government nor any agency thereof, nor any of their employees, makes any warranty, express or implied, or assumes any legal liability or responsibility for the accuracy, completeness, or usefulness of any information, apparatus, product, or process disclosed, or represents that its use would not infringe privately owned rights. Reference herein to any specific commercial product, process, or service by trade name, trademark, manufacturer, or otherwise does not necessarily constitute or imply its endorsement, recommendation, or favoring by the United States Government or any agency thereof. The views and opinions of authors expressed herein do not necessarily state or reflect those of the United States Government or any agency thereof.

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

# A    Appendix

## A.1    Details about Table 1

CP-ALS [13] is the standard, non-randomized alternating least squares method given by Algorithm 6 in Appendix A.9. The least squares problems in the algorithm are solved by exact methods. CP-ARLS-LEV is the algorithm proposed by Larsen and Kolda [15] that samples rows from the Khatri-Rao product according to a product distribution of leverage scores on each factor matrix. The per-iteration runtimes for both algorithms are re-derived in Appendix C.3 of the work by Malik [18] from their original sources. Malik [18] proposed the CP-ALS-ES algorithm (not listed in the table), which is superseded by the TNS-CP algorithm [19]. We report the complexity from Table 1 of the latter work. The algorithm by Ma and Solomonik [17] is based on a general method to sketch tensor networks. Our reported complexity is listed in Table 1 for Algorithm 1 in their work.

Table 1 does not list the one-time initialization costs for any of the methods. All methods require at least $O(NIR)$ time to randomly initialize factor matrices, and CP-ALS requires no further setup. CP-ARLS-LEV, TNS-CP, and STS-CP all require $O(NIR^2)$ initialization time. CP-ARLS-LEV uses the initialization phase to compute the initial leverage scores of all factor matrices. TNS-CP uses the initialization step to compute and cache Gram matrices of all factors $U_j$. STS-CP must build the efficient sampling data structure described in Theorem 1.1. The algorithm from Ma and Solomonik requires an initialization cost of $O(I^N m)$, where $m$ is a sketch size parameter that is $O(NR/\varepsilon^2)$ to achieve the $(\varepsilon, \delta)$ accuracy guarantee for each least squares solve.

## A.2    Definitions of Matrix Products

Table 2 defines the standard matrix product $\cdot$, Hadamard product $\circledast$, Kronecker product $\otimes$, and Khatri-Rao product $\odot$, as well as the dimensions of their operands.

Table 2: Matrix product definitions.

| OPERATION | SIZE OF $A$ | SIZE OF $B$ | SIZE OF $C$ | DEFINITION |
|---|---|---|---|---|
| $C = A \cdot B$ | $(m, k)$ | $(k, n)$ | $(m, n)$ | $C[i, j] = \sum_{a=1}^{k} A[i, a] B[a, j]$ |
| $C = A \circledast B$ | $(m, n)$ | $(m, n)$ | $(m, n)$ | $C[i, j] = A[i, j] B[i, j]$ |
| $C = A \otimes B$ | $(m_1, n_1)$ | $(m_2, n_2)$ | $(m_2 m_1, n_2 n_1)$ | $C[(i_2, i_1), (j_2, j_1)] = A[i_1, j_1] B[i_2, j_2]$ |
| $C = A \odot B$ | $(m_1, n)$ | $(m_2, n)$ | $(m_2 m_1, n)$ | $C[(i_2, i_1), j] = A[i_1, j] B[i_2, j]$ |

## A.3    Further Comparison to Prior Work

In this section, we provide a more detailed comparison of our sampling algorithm with the one proposed by Woodruff and Zandieh [29]. Their work introduces a ridge leverage-score sampling algorithm for Khatri-Rao products with the attractive property that the sketch can be formed in input-sparsity time. For constant failure probability $\delta$, the runtime to produce a $(1 \pm \epsilon)$ $\ell_2$-subspace embedding for $A = U_1 \odot ... \odot U_N$ is given in Appendix B of their work (proof of Theorem 2.7). Adapted to our notation, **their runtime** is

$$O\left( \log^4 R \log N \sum_{i=1}^{N} \text{nnz}(U_i) + \frac{N^7 s_\lambda^2 R}{\varepsilon^4} \log^5 R \log N \right)$$

where $s_\lambda = \sum_{i=1}^{R} \frac{\lambda_i}{\lambda_i + \lambda}$, $\lambda_1, ..., \lambda_R$ are the eigenvalues of the Gram matrix $G$ of matrix $A$, and $\lambda \geq 0$ is a regularization parameter. For comparison, **our runtime** for constant failure probability $\delta$ is

$$O\left( R \sum_{i=1}^{N} \text{nnz}(U_i) + \frac{R^3}{\varepsilon} \log\left( \prod_{i=1}^{N} I_i \right) \log R \right).$$

Woodruff and Zandieh's method provides a significant advantage for large column count $R$ or high regularization parameter $\lambda$. As a result, it is well-suited to the problem of regularized low-rank approximation when the column count $R$ is given by the number of data points in a dataset. On the

other hand, the algorithm has poor dependence on the matrix count $N$ and error parameter $\varepsilon$. For tensor decomposition, $R$ is typically no larger than a few hundred, while high accuracy ($\epsilon \approx 10^{-3}$) is required for certain tensors to achieve a fit competitive with non-randomized methods (see section 4.3, Figures 4 and 5). When $\lambda$ is small, we have $s_\lambda \approx R$. Here, Woodruff and Zandieh's runtime has an $O(R^3)$ dependence similar to ours. When $R \leq \log^4 R \log N$, our sampler has faster construction time as well.

Finally, we note that our sampling data structure can be constructed using highly cache-efficient, parallelizable symmetric rank-$R$ updates (BLAS3 operation `dSYRK`). As a result, the quadratic dependence on $R$ in our algorithm can be mitigated by dense linear algebra accelerators, such as GPUs or TPUs.

## A.4 Proof of Theorem 3.1

Theorem 3.1 appeared in a modified form as Lemma 10 in the work by Malik [18]. This original version used the definition $\tilde{G}_{>k} = \Phi \circledast \circledast_{a=k+1}^{N} G_k$ in place of $G_{>k}$ defined in Equation (4), where $\Phi$ was a sketched approximation of $G^+$. Woodruff and Zandieh [29] exhibit a version of the theorem with similar modifications. We prove the version stated in our work below.

*Proof of Theorem 3.1.* We rely on the assumption that the Khatri-Rao product $A$ is a nonzero matrix (but it may be rank-deficient). We begin by simplifying the expression for the leverage score of a row of $A$ corresponding to multi-index $(i_1, ..., i_N)$. Beginning with Equation (2), we derive

$$
\begin{aligned}
\ell_{i_1,...,i_N} &= A\left[(i_1,...,i_N),:\right] G^+ A\left[(i_1,...,i_N),:\right]^\top \\
&= \langle A\left[(i_1,...,i_N),:\right]^\top A\left[(i_1,...,i_N),:\right], G^+ \rangle \\
&= \langle \left( \circledast_{a=1}^{N} U_a\left[i_a,:\right] \right)^\top \left( \circledast_{a=1}^{N} U_a\left[i_a,:\right] \right), G^+ \rangle \\
&= \langle \circledast_{a=1}^{N} U_a\left[i_a,:\right]^\top U_a\left[i_a,:\right], G^+ \rangle \\
&= \langle \circledast_{a=1}^{k-1} U_a\left[i_a,:\right]^\top U_a\left[i_a,:\right], U_k\left[i_k,:\right]^\top U_k\left[i_k,:\right] \circledast \circledast_{a=k+1}^{N} U_a\left[i_a,:\right]^\top U_a\left[i_a,:\right], G^+ \rangle \\
&= \langle \circledast_{a=1}^{k-1} U_a\left[i_a,:\right]^\top U_a\left[i_a,:\right], U_k\left[i_k,:\right]^\top U_k\left[i_k,:\right], G^+ \circledast \circledast_{a=k+1}^{N} U_a\left[i_a,:\right]^\top U_a\left[i_a,:\right] \rangle.
\end{aligned}
\tag{13}
$$

We proceed to the main proof of the theorem. To compute $p(\hat{s}_k = s_k \mid \hat{s}_{<k} = s_{<k})$, we marginalize over random variables $\hat{s}_{k+1}...\hat{s}_N$. Recalling the definition of $h_{<k}$ from Equation (3), we have

$$
\begin{aligned}
p(\hat{s}_k = s_k \mid \hat{s}_{<k} = s_{<k}) &\propto \sum_{i_{k+1},...,i_N} p\left( (\hat{s}_{<k} = s_{<k}) \wedge (\hat{s}_k = s_k) \wedge \bigwedge_{u=k+1}^{N} (\hat{s}_u = i_u) \right) \\
&\propto \sum_{i_{k+1},...,i_N} \ell_{s_1,...,s_k,i_{k+1},...,i_N}.
\end{aligned}
\tag{14}
$$

The first line above follows by marginalizing over $\hat{s}_{k+1}, ..., \hat{s}_N$. The second line follows because the joint random variable $(\hat{s}_1, ..., \hat{s}_N)$ follows the distribution of statistical leverage scores on the rows of

$A$. We now plug in Equation (13) to get

$$\sum_{i_{k+1},\ldots,i_N} \ell_{s_1,\ldots,s_k,i_{k+1},\ldots,i_N}$$

$$= \sum_{i_{k+1},\ldots,i_N} \langle \left(\stackrel{k-1}{\underset{a=1}{\circledast}}\right) U_a\left[s_a,:\right]^\top U_a\left[s_a,:\right], U_k\left[s_k,:\right]^\top U_k\left[s_k,:\right], G^+ \circledast \left(\stackrel{N}{\underset{a=k+1}{\circledast}}\right) U_a\left[i_a,:\right]^\top U_a\left[i_a,:\right]\rangle$$

$$= \sum_{i_{k+1},\ldots,i_N} \langle h_{<k}h_{<k}^\top, U_k\left[s_k,:\right]^\top U_k\left[s_k,:\right], G^+ \circledast \left(\stackrel{N}{\underset{a=k+1}{\circledast}}\right) U_a\left[i_a,:\right]^\top U_a\left[i_a,:\right]\rangle$$

$$= \langle h_{<k}h_{<k}^\top, U_k\left[s_k,:\right]^\top U_k\left[s_k,:\right], G^+ \circledast \left(\stackrel{N}{\underset{a=k+1}{\circledast}}\right) \sum_{i_a=1}^{I_a} U_a\left[i_a,:\right]^\top U_a\left[i_a,:\right]\rangle$$

$$= \langle h_{<k}h_{<k}^\top, U_k\left[s_k,:\right]^\top U_k\left[s_k,:\right], G^+ \circledast \left(\stackrel{N}{\underset{a=k+1}{\circledast}}\right) G_a\rangle$$

$$= \langle h_{<k}h_{<k}^\top, U_k\left[s_k,:\right]^\top U_k\left[s_k,:\right], G_{>k}\rangle. \tag{15}$$

We now compute the normalization constant $C$ for the distribution by summing the last line of Equation (15) over all possible values for $\hat{s}_k$:

$$C = \sum_{s_k=1}^{I_k} \langle h_{<k}h_{<k}^\top, U_k\left[s_k,:\right]^\top U_k\left[s_k,:\right], G_{>k}\rangle$$

$$= \langle h_{<k}h_{<k}^\top, \sum_{s_k=1}^{I_k} U_k\left[s_k,:\right]^\top U_k\left[s_k,:\right], G_{>k}\rangle \tag{16}$$

$$= \langle h_{<k}h_{<k}^\top, G_k, G_{>k}\rangle.$$

For $k=1$, we have $h_{<k} = [1,\ldots,1]^\top$, so $C = \langle G_k, G_{>k}\rangle$. Then $C$ is the sum of all leverage scores, which is known to be the rank of $A$ [30]. Since $A$ was assumed nonzero, $C \neq 0$. For $k > 1$, assume that the conditioning event $\hat{s}_{<k} = s_{<k}$ occurs with nonzero probability. This is a reasonable assumption, since our sampling algorithm will never select prior values $\hat{s}_1,\ldots,\hat{s}_{k-1}$ that have 0 probability of occurrence. Let $\tilde{C}$ be the normalization constant for the conditional distribution on $\hat{s}_{k-1}$. Then we have

$$0 < p(\hat{s}_{k-1} = s_{k-1} \mid \hat{s}_{<k-1} = s_{<k-1})$$

$$= \tilde{C}^{-1} \langle h_{<k-1}h_{<k-1}^\top, U_{k-1}\left[s_{k-1},:\right]^\top U_{k-1}\left[s_{k-1},:\right], G_{>k-1}\rangle$$

$$= \tilde{C}^{-1} \langle h_{<k}h_{<k}^\top, G_{>k-1}\rangle \tag{17}$$

$$= \tilde{C}^{-1} \langle h_{<k}h_{<k}^\top, G_k \circledast G_{>k}\rangle$$

$$= \tilde{C}^{-1} \langle h_{<k}h_{<k}^\top, G_k, G_{>k}\rangle$$

$$= \tilde{C}^{-1} C$$

Since $\tilde{C} > 0$, we must have $C > 0$. $\qquad\square$

### A.5 Proof of Lemma 3.2

We detail the construction procedure, sampling procedure, and correctness of our proposed data structure. Recall that $T_{I,F}$ denotes the collection of nodes in a full, complete binary tree with $\lceil I/F \rceil$ leaves. Each leaf $v \in T_{I,F}$ holds a segment $S(v) = \{S_0(v)..S_1(v)\} \subseteq \{1..I\}$, with $|S(v)| \leq F$ and $S(u) \cap S(v) = \varnothing$ for distinct leaves $u, v$. For each internal node $v$, $S(v) = S(L(v)) \cup S(R(v))$, where $L(v)$ and $R(v)$ denote the left and right children of node $v$. The root node $r$ satisfies $S(r) = \{1..I\}$.

**Construction:** Algorithm 3 gives the procedure to build the data structure. We initialize a segment tree $T_{I,F}$ and compute $G^v$ for all leaf nodes $v \in T_{I,F}$ as a sum of outer products of rows from $U$ (lines 1-3). Starting at the level above the leaves, we then compute $G^v$ for each internal node as

the sum of $G^{L(v)}$ and $G^{R(v)}$, the partial Gram matrices of its two children. Runtime $O(IR^2)$ is required to compute $I$ outer products across all iterations of the loop on line 3. Our segment tree has $\lceil I/F \rceil - 1$ internal nodes, and the addition in line 6 contributes runtime $O(R^2)$ for each internal node. This adds complexity $O(R^2(\lceil I/F \rceil - 1)) \leq O(IR^2)$, for total construction time $O(IR^2)$.

To analyze the space complexity, observe that we store a matrix $G^v \in \mathbb{R}^{R \times R}$ at all $2\lceil I/F \rceil - 1$ nodes of the segment tree, for asymptotic space usage $O(\lceil I/F \rceil R^2)$. We can cut the space usage in half by only storing $G^v$ when $v$ is either the root or a left child in our tree, since the sampling procedure in Algorithm 4 only accesses the partial Gram matrix stored by left children. We can cut the space usage in half again by only storing the upper triangle of each symmetric matrix $G^v$. Finally, in the special case that $I < F$, the segment tree has depth 1 and the initial binary search can be eliminated entirely. As a result, the data structure has $O(1)$ space overhead, since we can avoid storing any partial Gram matrices $G^v$. This proves the complexity claims in point 1 of Lemma 3.2.

---

**Algorithm 3** BuildSampler($U \in \mathbb{R}^{I \times R}$, $F$, $Y$)

---

1: Build tree $T_{I,F}$ with depth $d = \lceil \log\lceil I/F \rceil \rceil$
2: **for** $v \in \text{leaves}(T_{I,F})$ **do**
3: $\quad G^v := \sum_{i \in S(v)} U[i,:]^\top U[i,:]$
4: **for** $u = d - 2...0$ **do**
5: $\quad$ **for** $v \in \text{level}(T_{I,F}, u)$ **do**
6: $\quad\quad G^v := G^{L(v)} + G^{R(v)}$

---

**Sampling:** Algorithm 4 gives the procedure to draw a sample from our proposed data structure. It is easy to verify that the normalization constant $C$ for $q_{h,U,Y}$ is $\langle hh^\top, G^{\text{root}(T_{I,F})}, Y \rangle$, since $G^{\text{root}(T_{I,F})} = U^\top U$. Lines 8 and 9 initialize a pair of templated procedures $\tilde{m}$ and $\tilde{q}$, each of which accepts a node from the segment tree. The former is used to compute the branching threshold at each internal node, and the latter returns the probability vector $q_{h,U,Y}[S_0(v) : S_1(v)]$ for the segment $\{S_0(v)..S_1(v)\}$ maintained by a leaf node. To see this last fact, observe for $i \in [I]$ that

$$
\begin{aligned}
&\tilde{q}(v)[i - S_0(v)] \\
&= C^{-1} U[i,:] \cdot (hh^\top \circledast Y) \cdot U[i,:]^\top \\
&= C^{-1} \langle hh^\top, U[i,:]^\top U[i,:], Y \rangle \\
&= q_{h,U,Y}[i].
\end{aligned}
\tag{18}
$$

The loop on line 12 performs the binary search using the two templated procedures. Line 18 uses the procedure $\tilde{q}$ to scan through at most $F$ bin endpoints after the binary search finishes early.

The depth of segment tree $T_{I,F}$ is $\log\lceil I/F \rceil$. As a result, the runtime of the sampling procedure is dominated by $\log\lceil I/F \rceil$ evaluations of $\tilde{m}$ and a single evaluation of $\tilde{q}$ during the binary search. Each execution of procedure $\tilde{m}$ requires time $O(R^2)$, relying on the partial Gram matrices $G^v$ computed during the construction phase. When $Y$ is a general p.s.d. matrix, the runtime of $\tilde{q}$ is $O(FR^2)$. This complexity is dominated by the matrix multiplication $W \cdot (hh^\top \circledast Y)$ on line 5. In this case, the runtime of the "RowSampler" procedure to draw one sample is $O(R^2 \log\lceil I/F \rceil + FR^2)$, satisfying the complexity claims in point 2 of the lemma.

Now suppppose $Y$ is a rank-1 matrix with $Y = uu^\top$ for some vector $u$. We have $hh^\top \circledast Y = (h \circledast u)(h \circledast u)^\top$. This gives

$$
\tilde{q}_p(h, C, v) = \text{diag}(W \cdot (hh^\top \circledast uu^\top) \cdot W) = (W \cdot (h \circledast u))^2
$$

where the square is elementwise. The runtime of the procedure $\tilde{q}$ is now dominated by a matrix-vector multiplication that costs time $O(FR)$. In this case, we have per-sample complexity $O(R^2 \log\lceil I/F \rceil + FR)$, matching the complexity claim in point 3 of the lemma.

**Correctness:** Recall that the inversion sampling procedure partitions the interval $[0, 1]$ into $I$ bins, the $i$-th bin having width $q_{h,U,Y}[i]$. The goal of our procedure is to find the bin that contains the uniform random draw $d$. Since procedure $\tilde{m}$ correctly returns the branching threshold (up to the offset "low") given by Equation (7), the loop on line 12 correctly implements a binary search on the list of bin endpoints specified by the vector $q_{h,U,Y}$. At the end of the loop, $c$ is a leaf node that

---

**Algorithm 4** Row Sampling Procedure

---

**Require:** Matrices $U, Y$ saved from construction, partial Gram matrices $\{G^v \mid v \in T_{I,F}\}$.

1: **procedure** $m_p(h, C, v)$
2:     **return** $C^{-1}\langle hh^\top, G^v, Y\rangle$
3: **procedure** $q_p(h, C, v)$
4:     $W := U[S(v), :]$
5:     **return** $C^{-1}\text{diag}(W \cdot (hh^\top \circledast Y) \cdot W^\top)$
6: **procedure** RowSample($h$)
7:     $C := \langle hh^\top, G^{\text{root}(T_{I,F})}, Y\rangle$
8:     $\tilde{m}(\cdot) := m_p(h, C, \cdot)$
9:     $\tilde{q}(\cdot) := q_p(h, C, \cdot)$
10:    $c := \text{root}(T_{I,F}), \text{low} = 0.0, \text{high} = 1.0$
11:    Sample $d \sim \text{Uniform}(0.0, 1.0)$
12:    **while** $c \notin \text{leaves}(T_{I,F})$ **do**
13:      $\text{cutoff} := \text{low} + \tilde{m}(L(c))$
14:      **if** cutoff $\geq d$ **then**
15:        $c := L(c), \text{high} := \text{cutoff}$
16:      **else**
17:        $c := R(c), \text{low} := \text{cutoff}$
18:    **return** $S_0(v) + \arg\min_{i \geq 0} \left( \text{low} + \sum_{j=1}^{i} \tilde{q}(c)[j] < d \right)$

---

maintains a collection $S(c)$ of bins, one of which contains the random draw $d$. Since the procedure $\tilde{q}$ correctly returns probabilities $q_{h,U,Y}[i]$ for $i \in S(c)$ for leaf node $c$, (see Equation (18)), line 18 finds the bin that contains the random draw $d$. The correctness of the procedure follows from the correctness of inversion sampling [22]. $\qquad\square$

## A.6 Cohesive Proof of Theorem 1.1

In this proof, we fully explain Algorithms 1 and 2 in the context of the sampling procedure outlined in section 3.2. We verify the complexity claims first and then prove correctness.

**Construction and Update:** For each matrix $U_j$, Algorithm 1 builds an efficient row sampling data structure $Z_j$ as specified by Lemma 3.2. We let the p.s.d. matrix $Y$ that parameterizes each sampler be a matrix of ones, and we set $F = R$. From Lemma 3.2, the time to construct sampler $Z_j$ is $O(I_j R^2)$. The space used by sampler $Z_j$ is $O(\lceil I_j/F \rceil R^2) = O(I_j R)$, since $F = R$. In case $I_j < R$, we use the special case described in Appendix A.5 to get a space overhead $O(1)$, avoiding a term $O(R^2)$ in the space complexity.

Summing the time and space complexities over all $j$ proves part 1 of the theorem. To update the data structure if matrix $U_j$ changes, we only need to rebuild sampler $Z_j$ for a cost of $O(I_j R^2)$. The construction phase also computes and stores the Gram matrix $G_j$ for each matrix $U_j$. We defer the update procedure in case a single entry of matrix $U_j$ changes to Appendix A.7.

**Sampling:** For all indices $k$ (except possibly $j$), lines 1-5 from Algorithm 2 compute $G_{>k}$ and its eigendecomposition. Only a single pass over the Gram matrices $G_k$ is needed, so these steps cost $O(R^3)$ for each index $k$. Line 5 builds an efficient row sampler $E_k$ for the matrix of scaled eigenvectors $\sqrt{\Lambda_k} \cdot V_k$. For sampler $k$, we set $Y = G_k$ with cutoff parameter $F = 1$. From Lemma 3.2, the construction cost is $O(R^3)$ for each index $k$, and the space required by each sampler is $O(R^3)$. Summing these quantities over all $k \neq j$ gives asymptotic runtime $O(NR^3)$ for lines 2-5.

The loop spanning lines 6-12 draws $J$ row indices from the Khatri-Rao product $U_{\neq j}$. For each sample, we maintain a "history vector" $h$ to write the variables $h_{<k}$ from Equation (3). For each index $k \neq j$, we draw random variable $\hat{u}_k$ using the row sampler $E_k$. This random draw indexes a scaled eigenvector of $G_{>k}$. We then use the history vector $h$ multiplied by the eigenvector to sample a row index $\hat{t}_k$ using data structure $Z_k$. The history vector $h$ is updated, and we proceed to draw the next index $\hat{t}_k$.

As written, lines 2-5 also incur scratch space usage $O(NR^3)$. The scratch space can be reduced to $O(R^3)$ by exchanging the order of loops on line 6 and line 8 and allocating $J$ separate history vectors $h$, once for each draw. Under this reordering, we perform all $J$ draws for each variable $\hat{u}_k$ and $\hat{t}_k$ before moving to $\hat{u}_{k+1}$ and $\hat{t}_{k+1}$. In this case, only a single data structure $E_k$ is required at each iteration of the outer loop, and we can avoid building all the structures in advance on line 5. We keep the algorithm in the form written for simplicity, but we implemented the memory-saving approach in our code.

From Lemma 3.2, lines 9 and 10 cost $O(R^2 \log R)$ and $O\left(R^2 \log \lceil I_k/R \rceil\right)$, respectively. Line 11 costs $O(R)$ and contributes a lower-order term. Summing over all $k \neq j$, the runtime to draw a single sample is

$$O\left(\sum_{k \neq j}(R^2 \log \lceil I_k/R \rceil + R^2 \log R)\right) = O\left(\sum_{k \neq j} R^2 \log \max\left(I_k, R\right)\right).$$

Adding the runtime for all $J$ samples to the runtime of the loop spanning lines 2-6 gives runtime $O\left(NR^3 + J\sum_{k \neq j} R^2 \log \max\left(I_k, R\right)\right)$, and the complexity claims have been proven.

**Correctness:** We show correctness for the case where $j = -1$ and we sample from the Khatri-Rao product of all matrices $U_k$, since the proof for any other value of $j$ requires a simple reindexing of matrices. To show that our sampler is correct, it is enough to prove the condition that for $1 \leq k \leq N$,

$$p(\hat{t}_k = t_k \mid h_{<k}) = q_{h_{<k}, U_k, G_{>k}}\left[t_k\right], \tag{19}$$

since, by Theorem 3.1, $p(\hat{s}_k = s_k \mid \hat{s}_{<k} = s_{<k}) = q_{h_{<k}, U_k, G_{>k}}\left[s_k\right]$. This would imply that the joint random variable $(\hat{t}_1, ..., \hat{t}_N)$ has the same probability distribution as $(\hat{s}_1, ..., \hat{s}_N)$, which by definition follows the leverage score distribution on $U_1 \odot ... \odot U_N$. To prove the condition in Equation (19), we apply Equations (9) and (11) derived earlier:

$$
\begin{aligned}
&p(\hat{t}_k = t_k \mid h_{<k}) \\
&= \sum_{u_k=1}^{R} p(\hat{t}_k = t_k \mid \hat{u}_k = u_k, h_{<k})p(\hat{u}_k = u_k \mid h_{<k}) && \text{(Bayes' Rule)} \\
&= \sum_{u_k=1}^{R} w\left[u_k\right] \frac{W\left[t_k, u_k\right]}{\|W\left[:, u_k\right]\|_1} && \text{(Equations (9) and (11), in reverse)} \\
&= q_{h_{<k}, U_k, G_{>k}}\left[t_k\right].
\end{aligned}
$$
$$\tag{20}$$
$\square$

## A.7 Efficient Single-Element Updates

Applications such as CP decomposition typically change all entries of a single matrix $U_j$ between iterations, incurring an update cost $O(I_j R^2)$ for our data structure from Theorem 1.1. In case only a single element of $U_j$ changes, our data structure can be updated in time $O\left(R \log I_j\right)$.

*Proof.* Algorithm 5 gives the procedure when the update $U_j\left[r, c\right] := \hat{u}$ is performed. The matrices $G^v$ refer to the partial Gram matrices maintained by each node $v$ of the segment trees in our data structure, and the matrix $\tilde{U}_j$ refers to the matrix $U_j$ before the update operation.

Let $T_{I_j, R}$ be the segment tree corresponding to matrix $U_j$ in the data structure, and let $v \in T_{I_j, R}$ be the leaf whose segment contains $r$. Lines 3-5 of the algorithm update the row and column indexed by $c$ in the partial Gram matrix held by the leaf node.

The only other nodes requiring an update are ancestors of $v$, each holding a partial Gram matrix that is the sum of its two children. Starting from the direct parent $A(v)$, the loop on line 6 performs these ancestor updates. The addition on line 8 only requires time $O(R)$, since only row and column $c$ change between the old value of $G^v$ and its updated version. Thus, the runtime of this procedure is $O(R \log \lceil I_j/R \rceil)$ from multiplying the cost to update a single node by the depth of the tree. $\square$

---

**Algorithm 5** UpdateSampler$(j, r, c, \hat{u})$

---

1: Let $u = \tilde{U}_j[r, c]$
2: Locate $v$ such that $r \in S(v)$
3: Update $G^v[c, :] \mathrel{+}= (\hat{u} - u)\tilde{U}_j[r, :]$
4: Update $G^v[:, c] \mathrel{+}= (\hat{u} - u)\tilde{U}_j[r, :]^\top$
5: Update $G^v[c, c] \mathrel{+}= (\hat{u} - u)^2$
6: **while** $v \neq \mathrm{root}(T_{I_j, R})$ **do**
7: $\quad v_{\mathrm{prev}} := v, v := A(v)$
8: $\quad$ Update $G^v := G^{\mathrm{sibling}(v_{\mathrm{prev}})} + G^{v_{\mathrm{prev}}}$

---

## A.8 Extension to Sparse Input Matrices

Our data structure is designed to sample from Khatri-Rao products $U_1 \odot ... \odot U_N$ where the input matrices $U_1, ..., U_N$ are dense, a typical situation in tensor decomposition. Slight modifications to the construction procedure permit our data structure to handle sparse matrices efficiently as well. The following corollary states the result as a modification to Theorem 1.1.

**Corollary A.1** (Sparse Input Modification). *When input matrices $U_1, ..., U_N$ are sparse, point 1 of Theorem 1.1 can be modified so that the proposed data structure has $O\left(R \sum_{j=1}^{N} nnz(U_j)\right)$ construction time and $O\left(\sum_{j=1}^{N} nnz(U_j)\right)$ storage space. The sampling time and scratch space usage in point 2 of Theorem 1.1 does not change. The single-element update time in point 1 is likewise unchanged.*

*Proof.* We will modify the data structure in Lemma 3.2. The changes to its construction and storage costs will propagate to our Khatri-Rao product sampler, which maintains one of these data structures for each input matrix.

Let us restrict ourselves to the case $F = R, Y = [1]$ in relation to the data structure in Lemma 3.2. These choices for $F$ and $Y$ are used in the construction phase given by Algorithm 1. The proof in Appendix A.5 constrains each leaf $v$ of a segment tree $T_{I,F}$ to hold a contiguous segment $S(v) \subseteq [I]$ of cardinality at most $F$. Instead, choose each segment $S(v) = \{S_0(v)..S_1(v)\}$ so that $U[S_0(v) : S_1(v), :]$ has at most $R^2$ nonzeros, and the leaf count of the tree is at most $\lceil nnz(U)/R^2\rceil + 1$ for input matrix $U \in \mathbb{R}^{I \times R}$. Assuming the nonzeros of $U$ are sorted in row-major order, we can construct such a partition of $[I]$ into segments in time $O(nnz(U))$ by iterating in order through the nonzero rows and adding each of them to a "current" segment. We shift to a new segment when the current segment cannot hold any more nonzeros.

This completes the modification to the data structure in Lemma 3.2, and we now analyze its updated time / space complexity.

**Updated Construction / Update Complexity of Lemma 3.2,** $F = R, Y = [1]$: Algorithm 3 constructs the partial Gram matrix for each leaf node $v$ in the segment tree. Each nonzero in the segment $U[S_0(v) : S_1(v), :]$ contributes time $O(R)$ during line 3 of Algorithm 3 to update a single row and column of $G^v$. Summed over all leaves, the cost of line 3 is $O(nnz(U)R)$. The remainder of the construction procedure updates the partial Gram matrices of all internal nodes. Since there are at most $O\left(\lceil nnz(U)/R^2\rceil\right)$ internal nodes and the addition on line 6 costs $O(R^2)$ per node, the remaining steps of the construction procedure cost $O(nnz(U))$, a lower-order term. The construction time is therefore $O(nnz(U)R)$.

Since we store a single partial Gram matrix of size $R^2$ at each of $O\left(\lceil nnz(U)/R^2\rceil\right)$ internal nodes, the space complexity of our modified data structure is $O(nnz(U))$.

Finally, the data structure update time in case a single element of $U$ is modified does not change from Theorem 1.1. Since the depth of the segment tree $\lceil nnz(U)/R^2\rceil + 1$ is upper-bounded by $\lceil I/R\rceil + 1$, the runtime of the update procedure in Algorithm 5 stays the same.

**Updated Sampling Complexity of Lemma 3.2,** $F = R, Y = [1]$: The procedure "RowSample" in Algorithm 4 now conducts a traversal of a tree of depth $O(\lceil nnz(U)/R^2\rceil)$. As a result, we

can still upper-bound the number of calls to procedure $\tilde{m}$ as $\lceil I/F \rceil$. The runtime of procedure $\tilde{m}$ is unchanged. The runtime of procedure $\tilde{q}$ for leaf node $c$ is dominated by the matrix-vector multiplication $U[S_0(c):S_1(c),:] \cdot h$. This runtime is $O(\text{nnz}(U[S_0(c):S_1(c),:])) \leq O(R^2)$. Putting these facts together, the sampling complexity of the data structure in Lemma 3.2 does not change under our proposed modifications for $F = R, Y = [1]$.

**Updated Construction Complexity of Theorem 1.1**: Algorithm 1 now requires $O\left(R \sum_{j=1}^{N} \text{nnz}(U_j)\right)$ construction time and $O\left(\sum_{j=1}^{N} \text{nnz}(U_j)\right)$ storage space, summing the costs for the updated structure from Lemma 3.2 over all matrices $U_1, ..., U_N$. The sampling complexity of these data structures is unaffected by the modifications, which completes the proof of the corollary. $\qquad\square$

## A.9 Alternating Least Squares CP Decomposition

**CP Decomposition.** CP decomposition represents an $N$-dimensional tensor $\tilde{\mathcal{T}} \in \mathbb{R}^{I_1 \times ... \times I_n}$ as a weighted sum of generalized outer products. Formally, let $U_1, ..., U_N$ with $U_j \in \mathbb{R}^{I_j \times R}$ be factor matrices with each column having unit norm, and let $\sigma \in \mathbb{R}^R$ be a nonnegative coefficient vector. We call $R$ the rank of the decomposition. The tensor $\tilde{\mathcal{T}}$ that the decomposition represents is given elementwise by

$$\tilde{\mathcal{T}}[i_1, ..., i_N] := \langle \sigma^\top, U_1[i_1,:], ..., U_N[i_N,:] \rangle = \sum_{r=1}^{R} \sigma[r] U_1[i_1, r] \cdots U_N[i_N, r],$$

which is a generalized inner product between $\sigma^\top$ and rows $U_j[i_j,:]$ for $1 \leq j \leq N$. Given an input tensor $\mathcal{T}$ and a target rank $R$, the goal of approximate CP decomposition is to find a rank-$R$ representation $\tilde{\mathcal{T}}$ that minimizes the Frobenius norm $\left\| \mathcal{T} - \tilde{\mathcal{T}} \right\|_F$.

**Definition of Matricization.** The matricization $\text{mat}(\mathcal{T}, j)$ flattens tensor $\mathcal{T} \in \mathbb{R}^{I_1 \times ... \times I_N}$ into a matrix and isolates mode $j$ along the row axis of the output. The output of matricization has dimensions $I_j \times \prod_{k \neq j} I_k$. We take the formal definition below from a survey by Kolda and Bader [13]. The tensor entry $\mathcal{T}[i_1, ..., i_N]$ is equal to the matricization entry $\text{mat}(\mathcal{T}, j)[i_N, u]$, where

$$u = 1 + \sum_{\substack{k=1 \\ k \neq j}}^{N} (i_k - 1) \prod_{\substack{m=1 \\ m \neq j}}^{k-1} I_m.$$

**Details about Alternating Least Squares.** Let $U_1, ..., U_N$ be factor matrices of a low-rank CP decomposition, $U_k \in \mathbb{R}^{I_k \times R}$. We use $U_{\neq j}$ to denote $\bigodot_{k=N, k \neq j}^{k=1} U_k$. Note the inversion of order here to match indexing in the definition of matricization above. Algorithm 6 gives the non-randomized alternating least squares algorithm CP-ALS that produces a decomposition of target rank $R$ given input tensor $\mathcal{T} \in \mathbb{R}^{I_1 \times ... \times I_N}$ in general format. The random initialization on line 1 of the algorithm can be implemented by drawing each entry of the factor matrices $U_j$ according to a standard normal distribution, or via a randomized range finder [11]. The vector $\sigma$ stores the generalized singular values of the decomposition. At iteration $j$ within a round, ALS holds all factor matrices except $U_j$ constant and solves a linear-least squares problem on line 6 for a new value for $U_j$. In between least squares solves, the algorithm renormalizes the columns of each matrix $U_j$ to unit norm and stores their original norms in the vector $\sigma$. Appendix A.11 contains more details about the randomized range finder and the convergence criteria used to halt iteration.

We obtain a randomized algorithm for sparse tensor CP decomposition by replacing the exact least squares solve on line 6 with a randomized method according to Theorem 2.1. Below, we prove Corollary 3.3, which derives the complexity of the randomized CP decomposition algorithm.

*Proof of Corollary 3.3.* The design matrix $U_{\neq j}$ for optimization problem $j$ within a round of ALS has dimensions $\prod_{k \neq j} I_k \times R$. The observation matrix $\text{mat}(\mathcal{T}, j)^\top$ has dimensions $\prod_{k \neq j} I_k \times I_j$. To achieve error threshold $1 + \varepsilon$ with probability $1 - \delta$ on each solve, we draw $J = \tilde{O}(R/(\varepsilon\delta))$ rows from both the design and observation matrices and solve the downsampled problem (Theorem 2.1).

---

**Algorithm 6** CP-ALS($\mathcal{T}$, $R$)

---

1: Initialize $U_j \in \mathbb{R}^{I_j \times R}$ randomly for $1 \le j \le N$.
2: Renormalize $U_j[:,i] \mathrel{/}= \|U_j[:,i]\|_2$, $1 \le j \le N, 1 \le i \le R$.
3: Initialize $\sigma \in \mathbb{R}^R$ to [1].
4: **while** not converged **do**
5: **for** $j = 1...N$ **do**
6:  $U_j := \arg\min_X \|U_{\ne j} \cdot X^\top - \mathrm{mat}(\mathcal{T}, j)^\top\|_F$
7:  $\sigma[i] = \|U_j[:,i]\|_2$, $1 \le i \le R$
8:  Renormalize $U_j[:,i] \mathrel{/}= \|U_j[:,i]\|_2$, $1 \le i \le R$.
9: **return** $[\sigma; U_1, ..., U_N]$.

---

These rows are sampled according to the leverage score distribution on the rows of $U_{\ne j}$, for which we use the data structure in Theorem 1.1. After a one-time initialization cost $O(\sum_{j=1}^{N} I_j R^2)$) before the ALS iteration begins, the complexity to draw $J$ samples (assuming $I_j \ge R$) is

$$
O\left( NR^3 + J\sum_{k \ne j} R^2 \log I_k \right) = \tilde{O}\left( NR^3 + \frac{R}{\varepsilon\delta}\sum_{k \ne j} R^2 \log I_k \right).
$$

The cost to assemble the corresponding subset of the observation matrix is $O(JI_j) = \tilde{O}(RI_j/(\varepsilon\delta))$. The cost to solve the downsampled least squares problem is $O(JR^2) = \tilde{O}(I_j R^2/(\varepsilon\delta))$, which dominates the cost of forming the subset of the observation matrix. Finally, we require additional time $O(I_j R^2)$ to update the sampling data structure (Theorem 1.1 part 1). Adding these terms together and summing over $1 \le j \le N$ gives

$$
\tilde{O}\left( \frac{1}{\varepsilon\delta} \cdot \sum_{j=1}^{N}\left[ I_j R^2 + \sum_{k \ne j} R^3 \log I_k \right] \right)
$$
$$
= \tilde{O}\left( \frac{1}{\varepsilon\delta} \cdot \sum_{j=1}^{N}\left[ I_j R^2 + (N-1)R^3 \log I_j \right] \right). \tag{21}
$$

Rounding $N-1$ to $N$ and multiplying by the number of iterations gives the desired complexity. When $I_j < R$ for any $j$, the complexity changes in Theorem 1.1 propagate to the equation above. The column renormalization on line 8 of the CP-ALS algorithm contributes additional time $O\left(\sum_{j=1}^{N} I_j R\right)$ per round, a lower-order term.

$\square$

## A.10 Experimental Platform and Sampler Parallelism

We provide two implementations of our sampler. The first is a slow reference implementation written entirely in Python, which closely mimics our pseudocode and can be used to test correctness. The second is an efficient implementation written in C++, parallelized in shared memory with OpenMP and Intel Thread Building Blocks.

Each Perlmutter CPU node (our experimental platform) is equipped with two sockets, each containing an AMD EPYC 7763 processor with 64 cores. All benchmarks were conducted with our efficient C++ implementation using 128 OpenMP threads. We link our code against Intel Thread Building blocks to call a multithreaded sort function when decomposing sparse tensors. We use OpenBLAS 0.3.21 to handle linear algebra with OpenMP parallelism enabled, but our code links against any linear algebra library implementing the CBLAS and LAPACKE interfaces.

Our proposed data structure samples from the exact distribution of leverage scores of the Khatri-Rao product, thereby enjoying better sample efficiency than alternative approaches such as CP-ARLS-LEV [15]. The cost to draw each sample, however, is $O(R^2 \log H)$, where $H$ is the number of rows in the Khatri-Rao product. Methods such as row-norm-squared sampling or CP-ARLS-LEV can draw each

sample in time $O(\log H)$ after appropriate preprocessing. Therefore, efficient parallelization of our sampling procedure is required for competitive performance, and we present two strategies below.

1. **Asynchronous Thread Parallelism**: The KRPSampleDraw procedure in Algorithm 2 can be called by multiple threads concurrently without data races. The simplest parallelization strategy divides the $J$ samples equally among the threads in a team, each of which makes calls to KRPSampleDraw asynchronously. This strategy works well on a CPU, but is less attractive on a SIMT processor like a GPU where instruction streams cannot diverge without significant performance penalties.

2. **Synchronous Batch Parallelism** As an alternative to the asynchronous strategy, suppose for the moment that all leaves have the same depth in each segment tree. Then for every sample, STSample makes a sequence of calls to $\tilde{m}$, each updating the current node by branching left or right in the tree. The length of this sequence is the depth of the tree, and it is followed by a single call to the function $\tilde{q}$. Observe that procedure $\tilde{m}$ in Algorithm 4 can be computed with a matrix-vector multiplication followed by a dot product. The procedure $\tilde{q}$ of Algorithm 4 requires the same two operations if $F = 1$ or $Y = [1]$. Thus, we can create a batched version of our sampling procedure that makes a fixed length sequence of calls to batched `gemv` and `dot` routines. All processors march in lock-step down the levels of each segment tree, each tracking the branching paths of a distinct set of samples. The MAGMA linear algebra library provides a batched version of `gemv` [10], while a batched dot product can be implemented with an ad hoc kernel. MAGMA also offers a batched version of the symmetric rank-$k$ update routine `syrk`, which is helpful to parallelize row sampler construction (Algorithm 3). When all leaves in the tree are not at the same level, the the bottom level of the tree can be handled with a special sequence of instructions making the required additional calls to $\tilde{m}$.

Our CPU code follows the batch synchronous design pattern. To avoid dependency on GPU-based MAGMA routines in our CPU prototype, portions of the code that should be batched BLAS calls are standard BLAS calls wrapped in a `for` loop. These sections can be easily replaced when the appropriate batched routines are available.

### A.11 Sparse Tensor CP Experimental Configuration

Table 3: Sparse Tensors from FROSTT collection.

| TENSOR | DIMENSIONS | NONZEROS | PREP. | INIT. |
|---|---|---|---|---|
| UBER PICKUPS | $183 \times 24 \times 1{,}140 \times 1{,}717$ | 3,309,490 | NONE | IID |
| ENRON EMAILS | $6{,}066 \times 5{,}699 \times 244{,}268 \times 1{,}176$ | 54,202,099 | LOG | RRF |
| NELL-2 | $12{,}092 \times 9{,}184 \times 28{,}818$ | 76,879,419 | LOG | IID |
| AMAZON REVIEWS | $4{,}821{,}207 \times 1{,}774{,}269 \times 1{,}805{,}187$ | 1,741,809,018 | NONE | IID |
| REDDIT-2015 | $8{,}211{,}298 \times 176{,}962 \times 8{,}116{,}559$ | 4,687,474,081 | LOG | IID |

Table 3 lists the nonzero counts and dimensions of sparse tensors in our experiments [25]. We took the log of all values in the Enron, NELL-2, and Reddit-2015 tensors. Consistent with established practice, this operation damps the effect of a few high magnitude tensor entries on the fit metric [15].

The factor matrices for the Uber, Amazon, NELL-2, and Reddit experiments were initialized with i.i.d. entries from the standard normal distribution. As suggested by Larsen and Kolda [15], the Enron tensor's factors were initialized with a randomized range finder [11]. The range finder algorithm initializes each factor matrix $U_j$ as $\text{mat}(\mathcal{T}, j)S$, a sketch applied to the mode-$j$ matricization of $\mathcal{T}$ with $S \in \mathbb{R}^{\prod_{k \neq j} I_k \times R}$. Larsen and Kolda chose $S$ as a sparse sampling matrix to select a random subset of fibers along each mode. We instead used an i.i.d. Gaussian sketching matrix that was not materialized explicitly. Instead, we exploited the sparsity of $\mathcal{T}$ and noted that at most $\text{nnz}(\mathcal{T})$ columns of $\text{mat}(\mathcal{T}, j)$ were nonzero. Thus, we computed at most $\text{nnz}(\mathcal{T})$ rows of the random sketching matrix $S$, which were lazily generated and discarded during the matrix multiplication without incurring excessive memory overhead.

ALS was run for a maximum of 40 rounds on all tensors except for Reddit, which was run for 80 rounds. The exact fit was computed every 5 rounds (defined as 1 epoch), and we used an early

stopping condition to terminate runs before the maximum round count. The algorithm was terminated at epoch $T$ if the maximum fit in the last 3 epochs did not exceed the maximum fit from epoch 1 through epoch $T - 3$ by tolerance $10^{-4}$.

Hybrid CP-ARLS-LEV deterministically includes rows from the Khatri-Rao product whose probabilities exceed a threshold $\tau$. The ostensible goal of this procedure is to improve diversity in sample selection, as CP-ARLS-LEV may suffer from many repeat draws of high probability rows. We replicated the conditions proposed in the original work by selecting $\tau = 1/J$ [15].

Individual trials of non-randomized (exact) ALS on the Amazon and Reddit tensors required several hours on a single Perlmutter CPU node. To speed up our experiments, accuracy measurements for exact ALS in Figure 3 were carried out using multi-node SPLATT, The Surprisingly ParalleL spArse Tensor Toolkit [24], on four Perlmutter CPU nodes. The fits computed by SPLATT agree with those computed by our own non-randomized ALS implementation. As a result, Figure 3 verifies that our randomized algorithm STS-CP produces tensor decompositions with accuracy comparable to those by highly-optimized, state-of-the-art CP decomposition software. We leave a distributed-memory implementation of our *randomized* algorithms to future work.

## A.12   Efficient Computation of Sketch Distortion

The definition of $\sigma$ in this section is different from its definition in the rest of this work. The condition number $\kappa$ of a matrix $M$ is defined as

$$\kappa(M) := \frac{\sigma_{\max}(M)}{\sigma_{\min}(M)}$$

where $\sigma_{\min}(M)$ and $\sigma_{\max}(M)$ denote the minimum and maximum nonzero singular values of $M$. Let $A$ be a Khatri-Rao product of $N$ matrices $U_1, ..., U_N$ with $\prod_{j=1}^N I_j$ rows, $R$ columns, and rank $r \le R$. Let $A = Q\Sigma V^\top$ be its reduced singular value decomposition with $Q \in \mathbb{R}^{\Pi_j I_j \times r}$, $\Sigma \in \mathbb{R}^{r \times r}$, and $V \in \mathbb{R}^{r \times R}$. Finally, let $S \in \mathbb{R}^{J \times \Pi_j I_j}$ be a leverage score sampling matrix for $A$. Our goal is to compute $\kappa(SQ)$ without fully materializing either $A$ or its QR decomposition. We derive

$$\begin{aligned} \kappa(SQ) &= \kappa(SQ\Sigma V^\top V \Sigma^{-1}) \\ &= \kappa(SAV\Sigma^{-1}) \end{aligned} \tag{22}$$

The matrix $SA \in \mathbb{R}^{J \times R}$ is efficiently computable using our leverage score sampling data structure. We require time $O(JR^2)$ to multiply it by $V\Sigma^{-1}$ and compute the singular value decomposition of the product to get the condition number. Next observe that $A^\top A = V\Sigma^2 V^\top$, so we can recover $V$ and $\Sigma^{-1}$ by eigendecomposition of $A^\top A \in \mathbb{R}^{R \times R}$ in time $O(R^3)$. Finally, recall the formula

$$A^\top A = \overset{N}{\underset{j=1}{\circledast}} U_j^\top U_j$$

used at the beginning of Section 3 that enables computation of $A^\top A$ in time $O\left(\sum_{j=1}^N I_j R^2\right)$ without materializing the full Khatri-Rao product. Excluding the time to form $SA$ (which is given by Theorem 1.1), $\kappa(SQ)$ is computable in time

$$O\left(JR^2 + R^3 + \sum_{j=1}^N I_j R^2\right).$$

Plugging $\kappa(SQ)$ into Equation (12) gives an efficient method to compute the distortion.

## A.13   Supplementary Results

### A.13.1   Comparison Against Standard CP Decomposition Packages

Table 4 compares the runtime per ALS round for our algorithm against existing common software packages for sparse tensor CP decomposition. We compared our algorithm against Tensorly version 0.81 [14] and Matlab Tensor Toolbox version 3.5 [3]. We compared our algorithm against both non-randomized ALS and a version of CP-ARLS-LEV in Tensor Toolbox.

Table 4: Average time (seconds) per ALS round for our method vs. standard CP decomposition packages. OOM indicates an out-of-memory error. All experiments were conducted on a single LBNL Perlmutter CPU node. Randomized algorithms were benchmarked with $2^{16}$ samples per least-squares solve.

| METHOD | UBER | ENRON | NELL-2 | AMAZON | REDDIT |
|---|---|---|---|---|---|
| TENSORLY, SPARSE BACKEND | 64.2 | OOM | 759.6 | OOM | OOM |
| MATLAB TTOOLBOX STANDARD | 11.6 | 294.4 | 177.4 | >3600 | OOM |
| MATLAB TTOOLBOX CP-ARLS-LEV | 0.5 | 1.4 | 1.9 | 34.2 | OOM |
| **STS-CP (ours)** | **0.2** | **0.5** | **0.6** | **3.4** | **26.0** |

As demonstrated by Table 4, our implementation exhibits more than 1000x speedup over Tensorly and 295x over Tensor Toolbox (non-randomized) for the NELL-2 tensor. STS-CP enjoys a dramatic speedup over Tensorly because the latter explicitly materializes the Khatri-Rao product, which is prohibitively expensive given the large tensor dimensions (see Table 3).

STS-CP consistently exhibits at least 2.5x speedup over the version of CP-ARLS-LEV in Tensor Toolbox, with more than 10x speedup on the Amazon tensor. To ensure a fair comparison with CP-ARLS-LEV, we wrote an improved implementation in C++ that was used for all other experiments.

### A.13.2 Probability Distribution Comparison

Figure 7 provides confirmation on a small test problem that our sampler works as expected. For the Khatri-Rao product of three matrices $A = U_1 \odot U_2 \odot U_3$, it plots the true distribution of leverage scores against a normalized histogram of 50,000 draws from the data structure in Theorem 1.1. We choose $U_1, U_2, U_3 \in \mathbb{R}^{8 \times 8}$ initialized i.i.d. from a standard normal distribution with 1% of all entries multiplied by 10. We observe excellent agreement between the histogram and the true distribution.

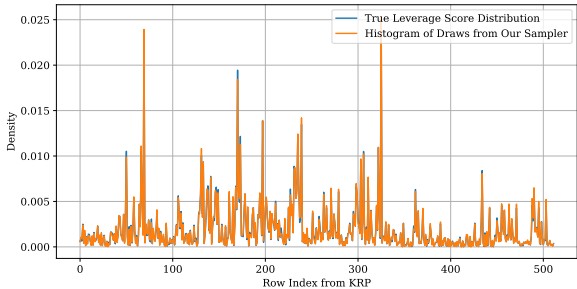

Figure 7: Comparison of true leverage score distribution with histogram of 50,000 samples drawn from $U_1 \odot U_2 \odot U_3$.

### A.13.3 Fits Achieved for $J = 2^{16}$

Table 5 gives the fits achieved for sparse tensor decomposition for varying rank and algorithm (presented graphically in Figure 4). Uncertainties are one standard deviation across 8 runs of ALS.

### A.13.4 Fit as a Function of Time

Figures 8a and 8b shows the fit as a function of time for the Amazon Reviews and NELL2 tensors. The hybrid version of CP-ARLS-LEV was used for comparison in both experiments. As in section 4.3, thick lines are averages of the running max fit across 4 ALS trials, shown by the thin dotted lines. For Amazon, the STS-CP algorithm makes faster progress than CP-ARLS-LEV at all tested sample counts.

For the NELL-2 tensor, STS-CP makes slower progress than CP-ARLS-LEV for sample counts up to $J = 163,840$. On average, these trials with CP-ARLS-LEV do not achieve the same final fit as STS-CP. CP-ARLS-LEV finally achieves a comparable fit to STS-CP when the former uses $J = 196,608$ samples, compared to $J = 65,536$ for our method.

Table 5: Fits Achieved by Randomized Algorithms for Sparse Tensor Decomposition, $J = 2^{16}$, and non-randomized ALS. The best result among randomized algorithms is bolded. "CP-ARLS-LEV-H" refers to the hybrid version of CP-ARLS-LEV and "Exact" refers to non-randomized ALS.

| TENSOR | $R$ | CP-ARLS-LEV | CP-ARLS-LEV-H | STS-CP (OURS) | EXACT |
|---|---|---|---|---|---|
| UBER | 25 | .187 ± 2.30E-03 | .188 ± 2.11E-03 | **.189** ± 1.52E-03 | .190 ± 1.41E-03 |
| | 50 | .211 ± 1.72E-03 | .212 ± 1.27E-03 | **.216** ± 1.18E-03 | .218 ± 1.61E-03 |
| | 75 | .218 ± 1.76E-03 | .218 ± 2.05E-03 | **.230** ± 9.24E-04 | .232 ± 9.29E-04 |
| | 100 | .217 ± 3.15E-03 | .217 ± 1.69E-03 | **.237** ± 2.12E-03 | .241 ± 1.00E-03 |
| | 125 | .213 ± 1.96E-03 | .213 ± 2.47E-03 | **.243** ± 1.78E-03 | .247 ± 1.52E-03 |
| ENRON | 25 | .0881 ± 1.02E-02 | .0882 ± 9.01E-03 | **.0955** ± 1.19E-02 | .0978 ± 8.50E-03 |
| | 50 | .0883 ± 1.72E-02 | .0920 ± 6.32E-03 | **.125** ± 1.03E-02 | .132 ± 1.51E-02 |
| | 75 | .0899 ± 6.10E-03 | .0885 ± 6.39E-03 | **.149** ± 1.25E-02 | .157 ± 4.87E-03 |
| | 100 | .0809 ± 1.26E-02 | .0787 ± 1.00E-02 | **.164** ± 5.90E-03 | .176 ± 4.12E-03 |
| | 125 | .0625 ± 1.52E-02 | .0652 ± 1.00E-02 | **.182** ± 1.04E-02 | .190 ± 4.35E-03 |
| NELL-2 | 25 | .0465 ± 9.52E-04 | .0467 ± 4.61E-04 | **.0470** ± 4.69E-04 | .0478 ± 7.20E-04 |
| | 50 | .0590 ± 5.33E-04 | .0593 ± 4.34E-04 | **.0608** ± 5.44E-04 | .0618 ± 4.21E-04 |
| | 75 | .0658 ± 6.84E-04 | .0660 ± 3.95E-04 | **.0694** ± 2.96E-04 | .0708 ± 3.11E-04 |
| | 100 | .0700 ± 4.91E-04 | .0704 ± 4.48E-04 | **.0760** ± 6.52E-04 | .0779 ± 5.09E-04 |
| | 125 | .0729 ± 8.56E-04 | .0733 ± 7.22E-04 | **.0814** ± 5.03E-04 | .0839 ± 8.47E-04 |
| AMAZON | 25 | .338 ± 6.63E-04 | .339 ± 6.99E-04 | **.340** ± 6.61E-04 | .340 ± 5.78E-04 |
| | 50 | .359 ± 1.09E-03 | .360 ± 8.04E-04 | **.366** ± 7.22E-04 | .366 ± 1.01E-03 |
| | 75 | .367 ± 1.82E-03 | .370 ± 1.74E-03 | **.382** ± 9.13E-04 | .382 ± 5.90E-04 |
| | 100 | .366 ± 3.05E-03 | .371 ± 2.53E-03 | **.392** ± 6.67E-04 | .393 ± 5.62E-04 |
| | 125 | .358 ± 6.51E-03 | .364 ± 4.22E-03 | **.400** ± 3.67E-04 | .401 ± 3.58E-04 |
| REDDIT | 25 | .0581 ± 1.02E-03 | .0583 ± 2.78E-04 | **.0592** ± 3.07E-04 | .0596 ± 4.27E-04 |
| | 50 | .0746 ± 1.03E-03 | .0738 ± 4.85E-03 | **.0774** ± 7.88E-04 | .0783 ± 2.60E-04 |
| | 75 | .0845 ± 1.64E-03 | .0849 ± 8.96E-04 | **.0909** ± 5.49E-04 | .0922 ± 3.69E-04 |
| | 100 | .0904 ± 1.35E-03 | .0911 ± 1.59E-03 | **.101** ± 6.25E-04 | .103 ± 7.14E-04 |
| | 125 | .0946 ± 2.13E-03 | .0945 ± 3.17E-03 | **.109** ± 7.71E-04 | .111 ± 7.98E-04 |

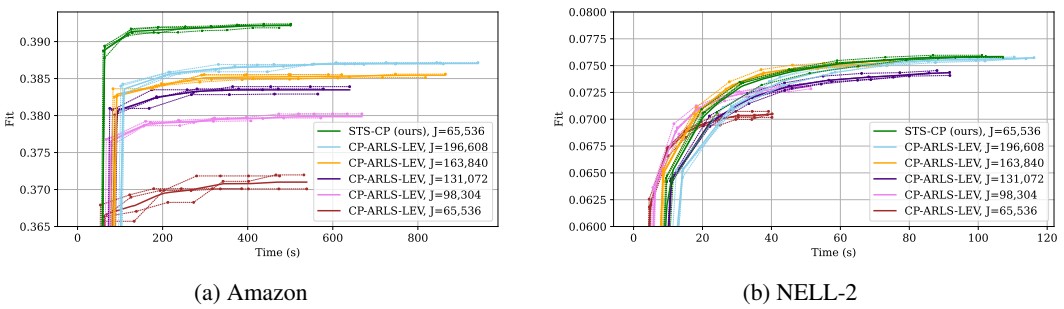

(a) Amazon    (b) NELL-2

Figure 8: Fit as a function of time, $R = 100$.

### A.13.5 Speedup of STS-CP and Practical Usage Guide

**Timing Comparisons.** For each tensor, we now compare hybrid CP-ARLS-LEV and STS-CP on the time required to achieve a fixed fraction of the fit achieved by non-randomized ALS. For each tensor and rank in the set $\{25, 50, 75, 100, 125\}$, we ran both algorithms using a range of sample counts. We tested STS-CP on values of $J$ from the set $\{2^{15}x \mid 1 \le x \le 4\}$ for all tensors. CP-ARLS-LEV required a sample count that varied significantly between datasets to hit the required thresholds, and we report the sample counts that we tested in Table 6. Because CP-ARLS-LEV has poorer sample complexity than STS-CP, we tested a wider range of sample counts for the former algorithm.

Table 6: Tested Sample Counts for hybrid CP-ARLS-LEV

| TENSOR | VALUES OF $J$ TESTED |
|---|---|
| UBER | $\{2^{15}x \mid x \in \{1..13\}\}$ |
| ENRON | $\{2^{15}x \mid x \in \{1..7\}\} \cup \{10, 12, 14, 16, 18, 20, 22, 26, 30, 34, 38, 42, 46, 50, 54\}\}$ |
| NELL-2 | $\{2^{15}x \mid x \in \{1..7\}\}$ |
| AMAZON | $\{2^{15}x \mid x \in \{1..7\}\}$ |
| REDDIT | $\{2^{15}x \mid x \in \{1..12\}\}$ |

For each configuration of tensor, target rank $R$, sampling algorithm, and sample count $J$, we ran 4 trials using the configuration and stopping criteria in Appendix A.11. The result of each trial was a set of (time, fit) pairs. For each configuration, we linearly interpolated the pairs for each trial and averaged the resulting continuous functions over all trials. The result for each configuration was a function $f_{\mathcal{T},R,A,J} : \mathbb{R}^+ \to [0,1]$. The value $f_{\mathcal{T},R,A,J}(t)$ is the average fit at time $t$ achieved by algorithm $A$ to decompose tensor $\mathcal{T}$ with target rank $R$ using $J$ samples per least squares solve. Finally, let

$$\text{Speedup}_{\mathcal{T},R,M} := \frac{\min_J \text{argmin}_{t \geq 0} \left[ f_{\mathcal{T},R,\text{CP-ARLS-LEV-H},J}(t) > P \right]}{\min_J \text{argmin}_{t \geq 0} \left[ f_{\mathcal{T},R,\text{STS-CP},J}(t) > P \right]}$$

be the speedup of STS-CP to over CP-ARLS-LEV (hybrid) to achieve a threshold fit $P$ on tensor $\mathcal{T}$ with target rank $R$. We let the threshold $P$ for each tensor $\mathcal{T}$ be a fixed fraction of the fit achieved by non-randomized ALS (see Table 5).

Figure 9 reports the speedup of STS-CP over hybrid CP-ARLS-LEV for $P = 0.95$ on all tensors except Enron. For large tensors with over one billion nonzeros, we report a significant speedup anywhere from 1.4x to 2.0x for all tested ranks. For smaller tensors with less than 100 million nonzeros, the lower cost of each least squares solve lessens the impact of the expensive, more accurate sample selection phase of STS-CP. Despite this, STS-CP performs comparably to CP-ARLS-LEV at most ranks, with significant slowdown only at rank 25 on the smallest tensor Uber.

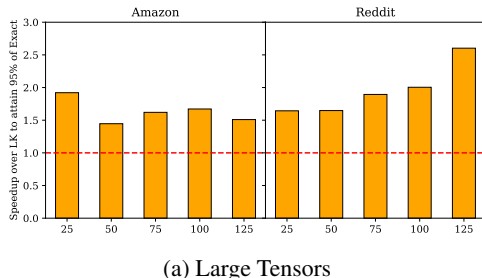

(a) Large Tensors

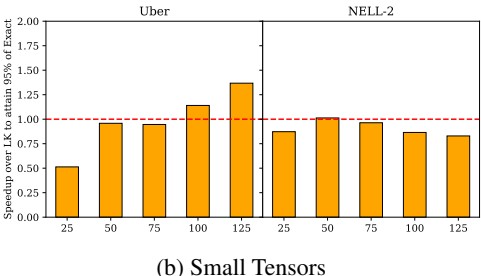

(b) Small Tensors

Figure 9: Speedup of STS-CP over CP-ARLS-LEV hybrid (LK) to reach 95% of the fit achieved by non-randomized ALS. Large tensors have more than 1 billion nonzero entries.

On the Enron tensor, hybrid CP-ARLS-LEV could not achieve the 95% accuracy threshold for any rank above 25 for the sample counts tested in Table 6. **STS-CP achieved the threshold accuracy for all ranks tested**. Instead, Figure 10 reports the speedup to achieve 85% of the fit of non-randomized ALS on the Enron. Beyond rank 25, our method consistently exhibits more than 2x speedup to reach the threshold.

**Guide to Sampler Selection.** Based on the performance comparisons in this section, we offer the following guide to CP decomposition algorithm selection. Our experiments demonstrate that **STS-CP offers the most benefit on sparse tensors with billions of nonzeros (Amazon and Reddit) at high target decomposition rank**. Here, the runtime of our more expensive sampling procedure is offset by reductions in the least squares solve time. For smaller tensors, our sampler may still offer significant performance benefits (Enron). In other cases (Uber, NELL-2), CP-ARLS-LEV exhibits better performance, but by small margins for rank beyond 50.

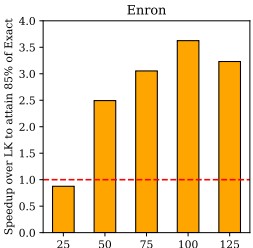

Figure 10: Speedup of STS-CP over CP-ARLS-LEV hybrid (LK) to reach 85% of the fit achieved by non-randomized ALS on the Enron Tensor.

STS-CP reduces the cost of each least squares solve through a sample selection process that relies on dense linear algebra primitives (see Algorithms 3 and 4). Because these operations can be expressed as standard BLAS calls and can be carried out in parallel (see Appendix A.10, we hypothesize that STS-CP is favorable when GPUs or other dense linear algebra accelerators are available.

Because our target tensor is sparse, the least squares solve during each ALS iteration requires a sparse matricized-tensor times Khatri-Rao product (spMTTKRP) operation. After sampling, this primitive can reduced to sparse-matrix dense-matrix multiplication (SpMM). Development of accelerators for these primitives is an active area of research [28, 26]. When such accelerators are available, the lower cost of the spMTTKRP operation reduces the relative benefit provided by the STS-CP sample selection method. We hypothesize that CP-ARLS-LEV, with its faster sample selection process but lower sample efficiency, may retain its benefit in this case. We leave verification of these two hypotheses as future work.

