# OpenReview forum: "Fast Exact Leverage Score Sampling from Khatri-Rao Products with Applications to Tensor Decomposition"
_NeurIPS.cc/2023/Conference — NeurIPS 2023 poster_

### Official Review · Reviewer_LxVj · 2023-06-17

**Soundness:** 3 good
**Presentation:** 3 good
**Contribution:** 3 good
**Rating:** 5
**Confidence:** 4

**Summary:**

This work studies fast, exact leverage score sampling for Khatri-Rao product
matrices (given access to its factor matrices) and an application to CP tensor
decomposition via alternating least squares with row sampling. The main
theoretical contribution is a data structure for sampling from the leverage
score distribution of a Khatri-Rao product by iteratively sampling from the
correct conditional marginal distribution as it constructs the row sample (one
factor at a time). To apply this most effectively to CP decompositions, the
authors propose a two-step version of this sampler that first samples a rank-1
eigenspace of the partial Gram factor matrices. The authors provide a good
comparison of their work to the leverage score-based CP decomposition algorithm
of Larsen and Kolda (Journal of Matrix Analysis and Applications, 2022).

**Strengths:**

- The sparse factor matrix version of Theorem 1.1 complements the paper.
- The comparison with Woodruff-Zandieh [27] is valuable and discusses the
  shortcomings of the theory-heavy results (i.e., hidden second-order constants
  that blow up).
- The paper is very nicely organized.
- The two-stage eigenspace sampling idea in Section 3.2 is a solid research
  contribution.

**Weaknesses:**

- Theorem 3.1 is one of the main theory components of the paper, but largely
  comes from [Malik, ICML 2022].
- Section 3.1 could benefit from a formal description of what the leaf nodes
  represent. In particular, what is $S_{0}(v)$ and how does this partitioning
  get decided (i.e., where to draw the cut points)? It seems reasonable that the
  result is correct, just not immediately implementable by the reader.
- In Corollary 3.3, it would be better to use the original expression $R=O(R
  \max\{\log(R/\delta), 1/(\varepsilon \delta)\})$ samples so as to not cause
  confusion by the implicit case assumption.
- Re experiments: It would be good to compare the running time of this
  sampling-based method with both ALS-ARLS-LEV and an out-of-the-box ALS
  implementation (e.g., MATLAB or Tensorly). Analyzing the fit of the
  decomposition in isolation doesn't tell the full story.

**Questions:**

**Questions**
- Re Theorem 1.1: If a single entry of $U_j$ changes, is there a faster update
  time?
- Should $I$ in Table 1 be $I_j$ to represent the $j$-th factor update?
- In Line 81, we have $S \in \mathbb{R}^{J \times I}$, so shouldn't we have $S[j,i]$ in
  the following line?
- In Line 95, what is the tilde in the lower bound for $J$? Related: Line 99
  should be $\Omega(R/(\varepsilon \delta))$ since this is a lower bound.
- In the discussion about Kronecker regression, Cheng et al. [4] and
  Larsen-Kolda [12] sample from the leverage score distribution of the
  Kronecker product for their CP decomposition algorithms (i.e., the product of
  the leverage scores). This connection seems worth pointing out.
- In Line 337, shouldn't the fixed sample count increase with the rank? This is
  needed for the leverage score sampling guarantees.

**Typos and suggestions**
- [line 14] "denoted" --> "denoted by"
- [line 43] Suggestion: Can remove the parenthesis around the summand in the
  second part of Theorem 1.1. The sentence "The structure can also draw
  samples..." can probably be generalized to say "... for any subset of factor
  matrices"?
- [line 49] You say that "our applications deal with dense inputs," but this is
  immediately followed by mention of the sparse Amazon tensor experiments. The
  writing here could be improved.
- [line 55] Suggestion: Add a citation for CP-ARLS-LEV after you first mention
  it.
- [line 58] Table 1 could benefit from $\tilde{O}$ notation to account for
  missing constant factors. The description of what the complexity is for could
  be made more explicit too: "Complexity of factor matrix $U_j$ for
  $N$-dimensional dense CP decomposition..."
- [line 64] suggestion: $j$'th --> $j$-th, same for later occurrences
- [line 79] suggestion: "sampling operators" --> "row sampling operators"
- [line 131] "Kronecker regression is distinct" --> "is a distinct"
- [line 141] suggestion: "autoregressive fashion" is an indirect way to explain
  the procedure. Consider dropping this phrase and merging the two sentences.
- [line 143] $I_n$ have been scalars so far. Therefore, it is better to use
  $(i_1, \dots, i_N) \in [I_1] \times \dots \times [I_N]$ to denote the set of
  indices.
- [line 144] suggestion: consider restating the dimension of $G_k$ and $G$ to
  help the reader remember what some of these operators mean, e.g.,
  $G := (\text{expression}) \in \mathbb{R}^{R \times R}$.
- [line 153] Please include the citation to Malik 2022 too, so that the reader
  can easily click through to the references.
- [line 162] Typo: "theorem 1.1" --> "Theorem 1.1"
- [line 165] Suggestion: Rewrite the sentence as: Let $h \in \mathbb{R}^{R}$ be
  a vector and let $Y \in \mathbb{R}^{R \times R}$ be a ... [delete
  "respectively"].

---

> ### Author Rebuttal · Authors · 2023-08-02
>
> We thank the reviewer, and we have already updated our draft to address the typos and suggestions. The table below gives the requested comparisons to out-of
> the-box methods. All times reported are average seconds
> per single ALS iteration; randomized algorithms were benchmarked
> with $2^{16}$ samples per LSTSQ solve. We used Python
> Tensorly 0.81 and Matlab Tensor Toolbox version
> 3.5 on an identical system
> configuration to other experiments. OOM
> denotes out-of-memory, for Tensorly due to explicit materialization
> of the Khatri-Rao product . Our method requires significantly
> less runtime across all tensors. Due to the slow
> performance of CP-ARLS-LEV in Matlab, we developed a
> high-performance multithreaded version in C++, which serves
> as the baseline in the rest of the paper. See figures 9 and 10 in the appendix (supplement to original submission)
> for more runtime comparisons.
>
> | Method    | Uber  | Enron | NELL-2 | Amazon | Reddit|
> | --------- | ----  | ----- | ------ | ------ | ----- |
> | Tensorly, Sparse Backend    | 64.2 | OOM  | 759.6 | OOM   | OOM    |
> | Matlab TToolbox Standard     | 11.6  | 249.4  | 177.4   | >3600 | OOM  |
> | Matlab TToolbox CP-ARLS-LEV  | 0.5 | 1.4 | 1.9  | 34.2  | OOM |
> | **STS-CP (Ours)** | 0.2 | 0.5 | 0.6  | 3.4  | 26.0  |
>
>
> Other Responses:
>
> **Questions:**
>
> 1. [Single Element Factor Updates] Yes! There is a faster update time, which is mentioned in Theorem 1.1 (line 41 of the submitted draft): “If a single entry in a matrix $U_j$ changes, it [the data structure] can be updated in time $O(R \log(\left| I_j \right| / R)$”. The procedure is detailed in Algorithm 5 in Appendix A.7, “Efficient Single-Element Updates”. The relatively simple method involves selectively updating matrices stored by binary tree nodes on the path from a leaf to the root, with the elements updated in each cached matrix depending on the index of the entry changed within $U_j$. Due to space constraints, we did not highlight this contribution more in the main body of the draft; we are happy to do so in the revision.
>
> 2. [Table 1 I vs I_j] We intended (but did not make clear in the caption) that the input tensor has dimensions $I \times I \times … \times I$ (i.e. all mode sizes $I_j = I$) to simplify the complexities. Thanks, we will clarify this.
>
> 3. [S[j, i] vs S[i, j] ] Corrected, thanks.
>
> 4. [Tilde on lower bound for J] The tilde denotes a hidden constant multiplying the right-hand side $R \max (\log (R / \delta), 1 / (\varepsilon \delta)))$. Woodruff [1, Theorem 2.11] reports the constant value as 144 multiplying the term $R \log (R / \delta)$, while Malik [2, S1 supplement] reports a value of $8 / 3$ as sufficient. Line 99: agreed, we have changed this to $\Omega$.
>
> 5. [Kronecker Sketching Connection] Thanks for pointing out this connection - we plan to include this in the section on Kronecker sketching.
>
> 6. [Increasing Sample Counts] We agree that the sample count should increase with the target rank in theory. However in figure 4, we would have to increase the sample count by different rates for different algorithms, since our algorithm STS-CP requires $O(R)$ samples, while CP-ARLS-LEV requires $O(R^{N-1})$ samples (see also Figure 6). Furthermore, our algorithm STS-CP performs better than worst-case analysis suggests when the sample budget is fixed, achieving 99.7% of the fit of exact ALS even at rank 125 on the Amazon tensor.
>
>    To avoid these confounding effects that vary between different algorithms and tensors, we used a fixed sample count throughout figure 4 to **directly compare the sample efficiency of our methods for a fixed budget**, as well as quantify the accuracy degradation as the rank increases. See figures 6, 8, 9, and 10 for experiments that vary the sample counts.
>
> **Weaknesses**
> 1. [Thereom 3.1 from Malik 2022]: You are correct. The novelty in our approach lies in the strategy to sample from the distribution given in Theorem 3.1, a critical improvement that enables scaling to massive sparse tensors with up to **several thousand times** less compute for decompositions of *identical* quality.
>
>    Suppose the algorithm in [Malik 2022] was applied to the Amazon Reviews tensor in our experiments with dimensions $4,821,207 \times 1,774,269 \times 1,805,187$ to produce a rank $R=25$ decomposition. After extra non-asymptotic improvements made in their paper, the exact floating point operation (FLOP) count to draw a sample from the KRP excluding the third mode is lower-bounded by $\left| I_2 \right| R^2$, or 1.12 gigaFLOPs *per row*. Including multiplicative constants but excluding the small $(25 \times 25)$-sized eigendecompositions (performed only per batch of rows, not once per row), **our** approach requires only 53.7 kiloFLOPs per row sample, a more than **20,000x reduction** that is due to the asymptotic improvement from $\left| I_j \right|$ to $\log \left| I_j \right|$.
>
>    We have revised our draft to emphasize these points.
>
> 2. [Interval Endpoints] Agreed, we revised our draft to make the cut points explicit. If we let $v_1, …, v_{\lceil I / F \rceil}$ be leaf nodes such that the intervals $S(v_1), …, S(v_N)$ are ordered from left to right, the explicit formula for $S_0(v_i)$ is
> $S_0(v_i) = (i-1) * F$, so that each segment has at most $F$ rows. Our draft has been updated to reflect this. The method to choose the cut points for sparse factors is slightly more involved, and the procedure is given in appendix A.8.
>
> 3. [Corollary 3.3] Corrected, thank you.
>
> 4. [Runtime Comparison] Agreed! Beyond Figures 6, 8a, and 8b, a thorough runtime comparison against CP-ARLS-LEV is made in Appendix A.12.4, figures 9 and 10, which measure the speedup of our algorithm over CP-ARLS-LEV. We tested a range of sample counts for both CP-ARLS-LEV and our algorithm, testing the former method on a significantly larger range of sample counts for fairness (see Table 5).

---

> > ### Comment · Reviewer_LxVj · 2023-08-13
> >
> > Thank you for the detailed response, especially the runtime comparison table.
> >
> > I also read all the other reviews and author responses, and am updating my rating to 5 (borderline accept).

---

### Official Review · Reviewer_qocR · 2023-06-29

**Soundness:** 4 excellent
**Presentation:** 4 excellent
**Contribution:** 3 good
**Rating:** 7
**Confidence:** 3

**Summary:**

The paper studies algorithms to efficiently sample a row from a matrix $A = U_1 \odot U_2 \odot ... \odot U_N$ with probability proportional to the leverage scores of $A$.
Here, $B \odot C$ denotes the _Khatri-Rao_ product of $B$ and $C$, or the column-wise kronecker-product of the columns of $B$ and $C$.
In particular, this means that $A$ has a small number of columns $R$, but has an exponential number of rows $\prod_{j=1}^N |I_j|$ (where $U_j \in \mathbb{R}^{|I_j| \times R}$).

This exponential blowup in the number of rows of $A$ in the core computational focus on this paper, as well as several related papers in the area.
This paper's technical contribution is in proposing a data structure that can preprocess $U_1, ..., U_N$ such that a row of $A$ can be sampled with probability proportional to its leverage score.
This data structure's space complexity matches the sizes of the input, and at query time it takes only $O(NR^2 \log \max \\{I_j, R\\})$ amortized time per row sampled.
This new data structure is based on "binary-tree inversion sampling", a binary search algorithm that can help sample from distributions efficiently, and is used to speed up a computational bottleneck in a known approach to sample leverage scores from Khatri-Rao products.

Theoretical results demonstrate the correctness of this algorithm, and experiments demonstrate the effectiveness of this algorithm.

**Strengths:**

The paper is well written, nicely motivated, pretty easy to understand, has a clear presentation of the subproblem it speeds up, and provides good context relative to existing work in the area (afaik; I'm no expert in tensor algorithms).

**Originality and Quality:** A fine line needs to be drawn here, and this is the subtlest point of my review. Per the authors' account, existing work in leverage score sampling from Khatri-Rao products exists and provides nice results. A good example is Theorem 3.1 from this paper, which is an adaptation of an observation made by prior work about a particular way to compute the leverage scores of $U_1 \odot ... \odot U_N$. The core of this paper is taking Theorem 3.1 and finding a faster way to compute it's right hand side. This means that the prior work gave much of the framework for around paper's proposed algorithm, and we can sorta think of this paper's contribution as finding a novel way to compute a subroutine more quickly.

That said, the approach to computing the subroutine is novel and interesting. I've not seen prior work in this area use a binary-tree in such a way. There's clearly a good deal of effort put into the math (though I only verified bits and pieces of the appendix). The paper very much stands as sufficiently original, but I want to be clear that the originality lands firmly in section 3.1 of the paper -- how to efficiently sample following this framework that existed in prior work.


**Clarity:** The paper is well written. I have only a couple mild gripes about writing and presentation (listed later in the "Questions" section), and I feel that I understood all the big ideas of this paper.


**Significance:** This paper feels well motivated, and the improvement to sample a row with runtime that depends logarithmically on the number of rows in $A$ is quiet strong and interesting. The experiments suggest that this truly is a state-of-the-art algorithm for a meaningful suite of metrics. I do have some gripes with the presentation of the experiments, which undercuts my confidence in the experimental evidence a smidge, but I think this both not essential and can be easily corrected.

**Weaknesses:**

The weaknesses are few and far between in this paper. A couple section lack a bit of clarity, like how the big-Oh rates for some of the downstream tensor algorithms are insufficiently explained. These are issues that can easily be fixed for a camera-ready version of the paper.

Out of these mild gripes, there are three that do stand out a bit:

1. Several experiments lack confidence intervals. This muddies the story of the algorithm's empirical efficacy, especially in Figure 3, where it feels very confusing to see the error of the green curve in the left figure jump up significantly.

2. In a few place, discussion of big-Oh runtime isn't fully explained. For instance, I don't understand how the complexity of STS-CP was derived on Table 1 (appendix A.1). More details are in the "Questions" section.

3. The last paragraph from section 3.2 takes the core technical ideas and is supposed to summarize the algorithm. However, it instead introduces some new notation and makes me wonder if it's actually summarizing an algorithm or if it's instead trying to briefly explain some of the deeper technical edgecases the algorithm has to handle. Either way, it's pretty confusing of a paragraph for me to read, and undercuts my confidence in understanding the full algorithm.

None of these are serious gripes, and call all be fixed with mild updates to the presentation. So I'm still very positive on this paper!

**Questions:**

I don't really have deep concerns to ask about here. I'll just enumerate a long list of minor typos and confusions.

1. Why use (afaik) non-standard notation, like having $d$ be the dimension (number of cols) of A, or $p_i$ be the probability of sampling row $i$, or $m$ or $c$ or $s$ being the number of sampled rows?
1. [Line 30] As someone familiar with leverage scores, I was surprised not to see a log appear in the sample complexity of leverage score sampling. I'd make it into a $\tilde O$.
1. [Lines 49 and 52] Line 49 says that the applications mainly deal with dense inputs. Line 52 says that the most practical benefit is on sparse matrices. These aren't necessarily statements that are at-odds, but it does feel like a bit of whiplash to read these statements back-to-back. The argument on lines 279-280 would be good to add here.
1. [Line 55] Do you have any understanding of why your algorithm which boasts a new and much smaller big-Oh term has 2% slower runtime?
1. [Table 1] I dunno how you came to the complexity-per-iteration here. As a reader, I felt like I should be able to pattern-match between Theorem 1.1 and Table 1 to understand the complexity of STS-CP, but I really couldn't make them line up.
1. [Line 131] "is **a** distinct"
1. [Line 132] "There" not "Here"
1. [Line 194] Get rid of the square on $[0,1]^2$
1. [Line 132] Explicitly argue why $F=1$ is affordable space-wise now, but was too expensive in the setting at the start of section 3.2
1. [Line 241] What is $Z_j$?
1. [Figures 2,3,5] Add confidence intervals
1. [Figure 4] Swap the x and y axis. It's hard to read with fit being on the x-axis. Add some space between the subplots -- they're real strange shoved together like that, making it harder to read.
1. [Line 344] Well... STS-CP does have an oscillating error pattern on Amazon too. Maybe mention it? Eh, I'm sorta torn here.
1. [Algo 6, line 5] Missing a period in $1,..,N$

---

> ### Author Rebuttal · Authors · 2023-08-04
>
> We appreciate your comments, which have strengthened the quality of our draft. The global rebuttal response to all reviewers includes an extra PDF that contains updated figures (2, 3, 4, 5) with error bars / axes swaps mentioned in this review. It also includes a frequency spectrum analysis (figure 6 in the extra PDF) regarding the error oscillation in CP-ARLS-LEV (see below).
>
> Responses:
>
> **Questions:**
> 1. [Notation] We chose the notation for column count [Larsen & Kolda 2022] and [Malik 2022], which use $R$ or $r$ as the column count (more standard in the tensor decomposition literature). We see your point with the other notation; happy to change $q_i$ to $p_i$. We were a bit worried that $s$ as the sample count could cause confusion with $\hat s_1, ..., \hat s_N$, denoting random variables for the multi-indices of a single sample, so we took $J$ as the sample count from [Malik 2022]. Open to changing this if necessary.
>
> 2. [O tilde] Agreed, and pointed out by another reviewer. We have updated our draft.
>
> 3. [Dense vs. Sparse Inputs] Agreed, thanks. We meant in lines 48-49 that the matrices $U_j$ in the Khatri-Rao product are typically dense, and will clarify this.
>
> 4. [2% Slower Runtime] This is because the sample counts for our algorithm vs CP-ARLS-LEV were equal for this first experiment (line 53), and CP-ARLS-LEV has faster runtime per sample but a poorer accuracy for the same sample budget, both in theory and practice.
>
>    In hindsight, this was not the best way to highlight our contributions, since the runtime, accuracy, and sample couns must all be accounted for. Our
>    revised draft makes reference to the last section of the appendix and reads as follows: "On the billion-scale Amazon and Reddit tensors,
>    our algorithm STS-CP can achieve 95% of the fit of exact ALS between 1.5x and 2.5x faster than the high-performance, state-of-
>    the-art competing CP-ARLS-LEV method. Our algorithm is significantly more sample-efficient; on the Enron tensor, only $2^{16}$ samples were required to achieve the 95%
>    accuracy threshold above a rank of 50, which could not be achieved by CP-ARLS-LEV with even 54 times as many samples.
>
> 5. [Complexity Derivation] per the comment above, we will work in $\tilde O$ notation. A version of the derivation below appears in appendix A.9, and we have added a reference to the main body of the text referencing it.
>
>    a. Each ALS iteration contains $N$ least-squares problems. The $j$'th problem, for $1 \leq j \leq N$, is $\min_X \left| \left| U_{\neq j} X - B \right| \right| $, where $U_{\neq j}$ is the Khatri-Rao product of all matrices but $U_j$. The matrix $B$ has $\left| I_j \right|$ columns.
>
>    b. For each problem, we need $\tilde O(R / (\epsilon \delta))$ samples from $A$ at a cost of $O(R^2 \sum_{k \neq j} \log \left| I_k \right|)$ per sample, plus a one-time cost (not-per-sample) of $O(NR^3)$ for the eigendecompositions. To avoid the dependence on $j$, we will round up the cost per sample to $O(R^2 \sum_{k=1}^N \log \left| I_k \right|)$
>
>    The cost to compute a $QR$ decomposition on the downsampled design matrix is $\tilde O(R^3 / (\epsilon \delta))$, a lower-order term. Because the observation matrix $B$ has $\left| I_j \right|$ columns, the cost of multiplying by $Q$ and back-substituting
>    with respect to $R$ is $\tilde O(\left| I_j \right| R^2 / (\epsilon \delta))$. The total complexity per solve is:
>
>    $\tilde O\left(NR^3 + (R / (\epsilon \delta)) \left(R^2 \sum_{k=1}^N \log \left| I_k \right| \right) + \left| I_j \right| R^2 / (\epsilon \delta)\right).$
>
>    For $\epsilon, \delta < 1$ and $\left|I_k\right| \geq 2$ for all $k$, observe that the first term $NR^3$ is at most the second term in the expression above, and we can eliminate it; simplifying slightly gives
>
>    $\tilde O\left((1 / (\epsilon \delta)) \left( \sum_{k=1}^N R^3 \log \left| I_k \right| \right) + \left| I_j \right| R^2 / (\epsilon \delta)\right)$
>
>    c. Finally, we sum the expression above over $1 \leq j \leq N$ to get:
>
>    $\tilde O\left((N / (\epsilon \delta)) \left( \sum_{k=1}^N R^3 \log \left| I_k \right| \right) + \sum_{j=1}^N \left| I_j \right| R^2 / (\epsilon \delta)\right)$
>
>    Combining the index variables $k$ and $j$ over independent sums into a single summation over $j$ gives:
>
>    $\tilde O\left((1 / (\epsilon \delta)) \sum_{j=1}^N \left( N R^3 \log \left| I_j \right| + \left| I_j \right| R^2 \right) \right)$
>
>    and setting all $\left| I_j \right|$ equal to $I$ gives the Table 1, last row.
>
> 6-9. Thanks, fixed.
>
> 10. The variables $Z_j$ are the segment trees (with cached matrices) for the second-stage eigensampling from each factor matrix, and the variables $E_k$ are the segment trees for the first stage involving the small $R \times R$ gram matrices. We clarified this and also revised the last paragraph of Section 3.2. See the global rebuttal for the revision text.
>
> 11, 12. Agreed, see PDF. We observe, in the updated Figure 3, that the median error of our method (orange bars) across several randomly generated input matrices actually decreases as the tensor dimension $N$ increases. On the other hand, there are more outliers (defined in the caption) that drive up the mean at increased dimension. We carefully checked that this behavior does not arise from experimental error. Both algorithms run on identical matrices regenerated once per new trial, with identical solve procedures after the sample indices are identified.
>
> 13. [Oscillating error pattern on Amazon] Our algorithm does exhibit some oscillation on Amazon, but we find no clearly-defined period. The period for CP-ARLS-LEV **exactly matches the dimension of the tensor**. This is an artifact that the worst-case theoretical guarantees for CP-ARLS-LEV do not capture. Figure 6 in the PDF attached is a frequency spectrum of the errors. Observe that CP-ARLS-LEV exhibits a clear peak at 4 for the Uber tensor and 3 for Amazon, while our algorithm does not suffer from such an artifact.
>
> 14. Thanks, corrected.

---

> > ### Comment · Reviewer_qocR · 2023-08-21
> > **Thanks for the response!**
> >
> > Sorry for the delay in my response. I enjoyed the authors' message, and happily maintain my score. The paper should be accepted.
> >
> > I'm glad to see the change in the figures, and the detailed responses in the bullet points above are well written. Some quick notes:
> >
> >  1 . If this is standard in parts of the tensor literature, then feel no need to change that notation. I'm not used to it, but that's a me problem.
> >
> >  4 . I really like that paragraph you inserted it. Maybe say ~65000 instead of $2^16$, just because $2^16$ feels like it should be much larger of a number than it actually is? Eh, kinda torn on this.
> >
> >  11 . I enjoy the updated figures, and they contain nice trend data. It's perfectly sufficient for a rebuttal, but for a camera-ready draft I think I'd recommend swapping the stars out -- they're hard to see unless I zoom in quite a bit.
> >
> >  13 . Strange, a good point from the frequency plot perspective. It might be worth acknowledging the distinction on line 345, but it's really up to you.

---

### Official Review · Reviewer_Njzt · 2023-07-01

**Soundness:** 3 good
**Presentation:** 2 fair
**Contribution:** 3 good
**Rating:** 7
**Confidence:** 4

**Summary:**

This work proposes a new algorithm to efficiently perform exact leverage score sampling on Khatri-Rao products. This work is built on top of the TNS-CP algorithm (Malik et al., 2022), and a more efficient data structure is used to achieve better sampling computational cost. This algorithm can be used to accelerate sketching based alternating least squares (ALS) for CP decomposition, and it has been shown that state-of-the-art complexity per ALS sweep has been achieved. Experimental results show that this sampler is efficient and yields better accuracy than previous leverage score sampling based CP-ALS algorithms for large real sparse tensors.



**Strengths:**

1. Accelerating leverage score sampling and large scale CP decomposition is an important topic, and this work achieves state-of-the-art results in both the theoretical analysis and the experimental results.

2. The data structure used to accelerate leverage score sampling of Khatri-Rao product is novel.

**Weaknesses:**

Overall I believe this is a good contribution. I only have one minor comment: it takes me a while to figure out the logics in Sections 3.1 and 3.2, and I think adding a figure to summarize the two sampling process would be good.



**Questions:**

n/a

---

> ### Author Rebuttal · Authors · 2023-08-02
>
> We appreciate your review of our paper. Per the instructions, the PDF attached to our global response at the top of these reviews has a three-part figure that illustrates the sampling process, including the matrices involved, the significant operations, and the two-stage sampling procedure. These diagrams have been added to our draft.

---

> > ### Comment · Reviewer_Njzt · 2023-08-19
> >
> > Thank you for the detailed response, I will keep my score.

---

### Official Review · Reviewer_u8zc · 2023-07-05

**Soundness:** 3 good
**Presentation:** 3 good
**Contribution:** 3 good
**Rating:** 5
**Confidence:** 3

**Summary:**

This paper designs a method for efficiently minimizing least squares regression, when the design matrix is equal to the Khatri-Rao product of multiple matrices. Because this Khatri-Rao product can have a huge dimension, the paper is motivated to study sampling techniques to reduce the dimension of the product matrix.

This least squares problem is motivated by applications to tensor decomposition, in particular when alternating least squares is used for finding CP tensor dimension. Each step of the alternating least squares corresponds to a least squares regression, where the design matrix is the Khatri-Rao product of the other tensor factors in the CP dimension.

To solve this sketching problem, the paper adapts the leverage score sampling technique to this Khatri-Rao product design matrix. The algorithm scales in a linear order proportionally to the sum of the dimensions of each matrix in the product.

Experiments are provided, comparing the approach to methods from a prior work by Larsen and Kolda (SIAM J. Matrix Analysis and Applications (2022)). The results show moderate improvement in terms of sparse tensor decomposition.

**Strengths:**

S1) The main is mainly of a technical nature: the algorithm, which is based on detailed calculations of the Khatri-Rao product, appears sound, and this is validated in the experiments.

S2) The sampling from Khatri-Rao product relies on a binary-tree inversion sampling technique, and this tree construction has time cost $O(I R^2)$ and storage cost $O(R^2 I / F)$.

S3) The experiments compared against the method of Larsen and Kolda appear significant, especially at higher target rank values, for several sparse tensors with as many as $10^9$ nonzeros.

**Weaknesses:**

W1) From my reading, the paper does not do a good job of motivating the problem. For instance, the main motivating example is tensor decomposition, but the connection is spelled explicitly only until Section 3.3. Are there any other plausible examples for motivating the proposed problem?

W2) I find the derivation steps to be quite laborious but also quite tedious. I find the illustration of Figure 1 to be very useful, and I think that a better presentation that conveys the key steps would be very helpful, for instance, in a special case of the Khatri-Rao product.

W3) The significance of the results is quite limited; in my opinion, I think the paper could benefit from stronger statements about how significant the contributions are. Solely judged based on the experiments, the improvement is quite weak since only one baseline is considered in the experiments. From a theoretical standpoint, the comparison stated in the related work could be made more clear as well.

**Questions:**

- It would be nice to give some description of the tensor used in the experiments (without having to refer to the appendix).
- I understand this question may be out of scope, but it would be nice to compare fit against other tensor decompositions, e.g., the Tucker decomposition. Would the newly developed techniques apply to Tucker decomposition?
- It is mentioned in the related work that this paper is very closely related to a method by Woodruff and Zandieh (2022). I wonder how the methods would compare against each other in the experiments.

**Limitations:**

Yes, the paper discussed limitations in Section 5. Due to the technical nature, I do not see any potential societal impact of their work.

---

> ### Author Rebuttal · Authors · 2023-08-02
>
> We thank the reviewer for their comments and useful feedback, which have strengthened the quality of our draft.
>
> Regarding the third weakness [W3] about the novelty of our contribution: our approach offers a significant advance in the best theoretical complexity for sketched ALS CP Decomposition. By improving the per-leverage-score sample complexity from linear in $\left| I_j \right|$ (Malik 2022) to $\log \left| I_j \right|$, we enable computational cost savings of **several thousand times** for approximations of **identical** quality. This is the critical improvement that allows exact leverage-score sketching for large sparse tensor decomposition, which we illustrate by the FLOP complexity calculation below.
>
>    Suppose the algorithm in [Malik 2022] was applied to the Amazon Reviews tensor in our experiments with dimensions $4,821,207 \times 1,774,269 \times 1,805,187$ to produce a rank $R=25$ decomposition. After extra non-asymptotic improvements made in their paper, the exact floating point operation (FLOP) count to draw a sample from the KRP excluding the third mode is lower-bounded by $\left| I_2 \right| R^2$, or 1.12 gigaFLOPs *per row*. Including multiplicative constants but excluding the small $(25 \times 25)$-sized eigendecompositions, **our** approach requires only 53.7 kiloFLOPs per row sample, a more than **20,000x reduction** that is due to the asymptotic improvement from $\left| I_j \right|$ to $\log \left| I_j \right|$.
>
> We plan to add the statements above to the paper to highlight our contributions over existing work.
>
> Other Responses:
>
> **Questions**
> 1. [Tensor Descriptions] Agreed. We omitted these descriptions due to space constraints in the main body, but have added them in the extra space the revision affords.
>
> 2. [Tucker Decomposition] The techniques developed here are less applicable to ALS Tucker decomposition due to the different properties of the Kronecker product on which it is based. The leverage scores on the rows of $A \otimes B$ are products of the leverage scores of $A$ and $B$, a property that the Khatri-Rao product does not share. Drawing a sample from the Kronecker product, therefore, is simpler than sampling from a Khatri-Rao product, although the number of samples required to achieve least-squares solution guarantees is exponential in the tensor dimension for even state-of-the-art sketched ALS Tucker methods (see, e.g. https://arxiv.org/abs/2209.04876). As a result, the runtime to compute the Tucker decomposition may be significantly higher than CP decomposition for even 4-dimensional tensors.
>
> 3. [Woodruff Zandieh Comparison] The algorithm developed by Woodruff and Zandieh is intricate (requiring three distinct sketching operators composed in non-trivial ways) with no publicly available code, to the best of our knowledge. Furthermore, their near input-sparsity time algorithm confers no benefit for tensor decomposition. The least-squares problems $\min_X \left| \left| AX - B \right| \right|$ in CP decomposition have observation matrices $B$ with $\left| I_j \right|$ columns for each $1 \leq j \leq N$. Regardless of the efficiency of the sketching mechanism, the column count of matrix $B$ introduces an unavoidable cost $O(\left| I_j \right| R^2)$, destroying the benefit of the $\tilde O(\left| I_j \right| R)$ input-sparsity time cost to form the sketching data structure for each factor matrix $U_j$. Furthermore, Woodruff and Zandieh do not provide a readily evident method to update their sketch when entries of each factor matrix change, although their approach could likely be adapted. Efficient factor matrix updates are essential for CP decomposition, and our algorithm provides methods to update each factor when even a single entry changes.
>
>
> **Weaknesses**
>
> 1. [W1] Our introduction includes several motivating examples for linear systems with Khatri-Rao design matrices, including compressed sensing and signal processing besides tensor decomposition. Systems of this exact form also occur in PDE-inverse problems (see, for example, equations (2) and (4) from https://arxiv.org/pdf/1909.11290.pdf); like us, they also consider the case with low column count and hundreds of thousands of rows per matrix forming the Khatri-Rao product. We have added this reference to the paper.
>
> 2. [W2] In the global response at the top of these reviews, the PDF attached contains figures  that illustrate Theorem 3.1 and our two-stage leverage-score sampling procedure. We have added these to our draft. We hope these aid the reader and are happy to engage in further discussion.
>
> 3. [W3] The main baseline that we selected for comparison, CP-ARLS-LEV, is a state-of-the-art algorithm for sparse tensor decomposition with performance exceeding well-known software packages on the market. The table below compares the runtime per iteration of our algorithm with three other pieces of software for sparse tensor decomposition. Figure 4 (original document) and Table 4 demonstrate that our accuracy is comparable to non-sketched tensor decomposition, recovering 99.7% of the fit consistently for the Amazon tensor.
>
> | Method    | Uber  | Enron | NELL-2 | Amazon | Reddit|
> | - | - | - | - | - | - |
> | Tensorly, Sparse Backend | 64.2 | OOM  | 759.6 | OOM   | OOM    |
> | Matlab TToolbox Standard | 11.6  | 249.4  | 177.4   | >3600 | OOM |
> | Matlab TToolbox CP-ARLS-LEV | 0.5 | 1.4 | 1.9  | 34.2  | OOM|
> | **STS-CP (Ours)** | 0.2 | 0.5 | 0.6  | 3.4  | 26.0 |
>
> All times reported are average seconds
> per single ALS iteration; randomized algorithms were benchmarked
> with $2^{16}$ samples per LSTSQ solve. We used Python
> Tensorly 0.81 and Matlab Tensor Toolbox version
> 3.5 on an identical system
> configuration to other experiments. OOM means out-of-memory. **The timings demonstrate that our algorithm STS-CP can quickly decompose tensors requiring hundreds of gigabytes of disk space in a fraction of the time required by standard packages**. We have added these clarifications to a revised copy of our draft.

---

### Author Rebuttal · Authors · 2023-08-08

We thank all the reviewers for their constructive comments. The PDF attached to this rebuttal contains figures that were requested by some reviewers. These include diagrams that illustrate our sampling procedure and updated graphs with error bars. Below, you can find a summary of some revisions to our draft common to multiple reviewers, with specific responses in the individual rebuttals to each review.

1. We revised our draft to use $\tilde O$ notation instead of $O$ notation, where applicable, to account for the factors $\log(1 / \delta)$ that appear in leverage-score sampling.

2. We provided diagrams illustrating Theorem 3.1 and the two-stage sampling process, Figure 1 in the attached PDF.

3. We have provided baseline comparisons to state-of-the-art tensor decomposition libraries, such as Tensorly and Tensortoolbox, that demonstrate the dramatic efficiency of our algorithm while maintaining approximation quality. The responses to reviewers u8zc and LxVj contain this table with average runtime per ALS iteration for our method vs. Tensorly and Matlab TensorToolbox 3.5.

We hope that these changes improve the clarity of the draft, and are happy to engage in further discussion.

Finally, reviewer qocR requested a revision of the last paragraph of section 3.2, which we include here due to space constraints in the comment below that review. Point taken, this last paragraph is dense and can be simplified. Here is our revision of that paragraph:

"To summarize, Algorithms 1 and 2 give the construction procedure and two-stage sampling algorithm described above. The subroutines "BuildSampler" and "RowSample" relate to the procedure to build and sample, respectively, from the data structure in Lemma 3.2. The construction procedure builds the samplers $Z_j$, $1 \leq j \leq N$, for the second phase of sampling. The construction cost is $O(\left| I_j  \right| R^2)$ per matrix $U_j$. The sampling algorithm returns $J$ samples from the Khatri-Rao product of all matrices (excluding possibly index $U_j$, a useful feature for tensor decomposition applications). Lines 2-5 construct the sampling data structures for the first phase of sampling, while lines 9-11 implement the two-stage procedure by calling RowSample twice in succession and updating the running vector $h_{<k}$."

---

### Decision · Program_Chairs · 2023-09-21

**Decision:**

Accept (poster)

**Comment:**

Dear Authors,

Thank you for your valuable contribution to Neurips and the ML community. Your submitted paper has undergone a rigorous review process, and I have carefully read feedback provided by the reviewers and considered the author rebuttal in detail.

This paper proposes a data structure that enables sampling rows from a Khatri-Rao product with respect to leverage scores. The key idea is using a "binary-tree inversion sampling" strategy. Since such matrices arise in least squares problems arising in tensor decompositions, the proposed method can be used in accelerating approximate solutions of such problems. The reviewers all agree that the contribution is solid and interesting.

Given this positive assessment, I am willing to recommend the acceptance of your paper for publication.

I would like to remind you to carefully review the reviewer feedback and the resulting discussion. While most reviews were positive, the reviewers have offered valuable suggestions that can further strengthen the quality of the paper. Please take another careful look a the 'weaknesses' section of each reviewer comment. Please also review the discussions with Reviewers LxVj and u8zc in order to resolve any potential confusion. I encourage you to use this feedback to make any necessary improvements and refinements before submitting the final version of your paper.

Once again, thank you for submitting your work to Neurips.

Best,
Area Chair